# A Best-of-Both-Worlds Algorithm for Bandits with Delayed Feedback

**Saeed Masoudian**
University of Copenhagen
saeed.masoudian@di.ku.dk

**Julian Zimmert**
Google Research
zimmert@google.com

**Yevgeny Seldin**
University of Copenhagen
seldin@di.ku.dk

## Abstract

We present a modified tuning of the algorithm of Zimmert and Seldin [2020] for adversarial multiarmed bandits with delayed feedback, which in addition to the minimax optimal adversarial regret guarantee shown by Zimmert and Seldin simultaneously achieves a near-optimal regret guarantee in the stochastic setting with fixed delays. Specifically, the adversarial regret guarantee is $\mathcal{O}(\sqrt{TK} + \sqrt{dT \log K})$, where $T$ is the time horizon, $K$ is the number of arms, and $d$ is the fixed delay, whereas the stochastic regret guarantee is $\mathcal{O}\left(\sum_{i \neq i^*}(\frac{1}{\Delta_i} \log(T) + \frac{d}{\Delta_i \log K}) + dK^{1/3} \log K\right)$, where $\Delta_i$ are the suboptimality gaps. We also present an extension of the algorithm to the case of arbitrary delays, which is based on an oracle knowledge of the maximal delay $d_{max}$ and achieves $\mathcal{O}(\sqrt{TK} + \sqrt{D \log K} + d_{max}K^{1/3} \log K)$ regret in the adversarial regime, where $D$ is the total delay, and $\mathcal{O}\left(\sum_{i \neq i^*}(\frac{1}{\Delta_i} \log(T) + \frac{\sigma_{max}}{\Delta_i \log K}) + d_{max}K^{1/3} \log K\right)$ regret in the stochastic regime, where $\sigma_{max}$ is the maximal number of outstanding observations. Finally, we present a lower bound that matches the refined adversarial regret upper bound achieved by the skipping technique of Zimmert and Seldin [2020] in the adversarial setting.

## 1 Introduction

Delayed feedback is a common challenge in many online learning problems, including multi-armed bandits. The literature studying multi-armed bandit games with delayed feedback builds on prior work on bandit problems with no delays. The researchers have traditionally separated the study of bandit games in stochastic environments [Thompson, 1933, Robbins, 1952, Lai and Robbins, 1985, Auer et al., 2002] and in adversarial environments[Auer et al., 2002b]. However, in practice the environments are rarely purely stochastic, whereas they may not be fully adversarial either. Furthermore, the exact nature of an environment is not always known in practice. Therefore, in recent years there has been an increasing interest in algorithms that perform well in both regimes with no prior knowledge of the regime [Bubeck and Slivkins, 2012, Seldin and Slivkins, 2014, Auer and Chiang, 2016, Seldin and Lugosi, 2017, Wei and Luo, 2018]. The quest for best-of-both-worlds algorithms for no-delay setting culminated with the Tsallis-INF algorithm proposed by Zimmert and Seldin [2019], which achieves the optimal regret bounds in both stochastic and adversarial environments. The algorithm and analysis were further improved by Zimmert and Seldin [2021] and Masoudian and Seldin [2021], who, in particular, derived improved regret bounds for intermediate

regimes between stochastic and adversarial, while Ito [2021] removed an assumption on uniqueness of the best arm, which was used in the early works.

Our goal is to extend best-of-both-worlds results to multi-armed bandits with delayed feedback. So far the literature on multi-armed bandits with delayed feedback has followed the traditional separation into stochastic and adversarial. In the stochastic regime Joulani et al. [2013] showed that if the delays are random (generated i.i.d), then compared to the non-delayed stochastic multi-armed bandit setting, the regret only increases additively by a factor that is proportional to the expected delay. In the adversarial setting Cesa-Bianchi et al. [2019] have studied the case of uniform delays $d$. They derived a lower bound $\Omega(\max(\sqrt{KT}, \sqrt{dT \log K}))$ and an almost matching upper bound $\mathcal{O}(\sqrt{KT \log K} + \sqrt{dT \log K})$. Thune et al. [2019] and Bistritz et al. [2019] extended the results to arbitrary delays, achieving $\mathcal{O}(\sqrt{KT \log K} + \sqrt{D \log K})$ regret bounds based on oracle knowledge of the total delay $D$ and time horizon $T$. Thune et al. [2019] also proposed a skipping technique based on advance knowledge of the delays "at action time", which allowed to exclude excessively large delays from $D$. Finally, Zimmert and Seldin [2020] introduced an FTRL algorithm with a hybrid regularizer that achieved $\mathcal{O}(\sqrt{KT} + \sqrt{D \log K})$ regret bound, matching the lower bound in the case of uniform delays and requiring no prior knowledge of $D$ or $T$. The regularizer used by Zimmert and Seldin was a mix of the negative Tsallis entropy regularizer used in the Tsallis-INF algorithm for bandits and the negative entropy regularizer used in the Hedge algorithm for full information games, mixed with separate learning rates:

$$F_t(x) = -2\eta_t^{-1} \left( \sum_{i=1}^{K} \sqrt{x_i} \right) + \gamma_t^{-1} \left( \sum_{i=1}^{K} x_i(\log x_i - 1) \right). \tag{1}$$

Zimmert and Seldin [2020] also improved the skipping technique and achieved a refined regret bound $\mathcal{O}(\sqrt{KT} + \min_S(|S| + \sqrt{D_{\bar{S}} \log K}))$, where $S$ is a set of skipped rounds and $D_{\bar{S}}$ is the total delay in non-skipped rounds. The refined skipping technique requires no advance knowledge of the delays. Their key step toward elimination of the need of advance knowledge of delays was to base the analysis on the count of the number of outstanding observations rather than the delays. The great advantage of skipping is that a few rounds with excessively large or potentially even infinite delays have a very limited impact on the regret bound. One of our contributions in this paper is a lower bound for the case of non-uniform delays, which matches the refined regret upper bound achieved by skipping.

Even though the hybrid regularizer used by Zimmert and Seldin [2020] was sharing the Tsallis entropy part with their best-of-both-worlds Tsallis-INF algorithm from Zimmert and Seldin [2021], and even though the adversarial analysis was partly similar to the analysis of the Tsallis-INF algorithm, Zimmert and Seldin [2020] did not manage to derive a regret bound for their algorithm in the stochastic setting with delayed feedback and left it as an open problem. The stochastic analysis of the Tsallis-INF algorithm is based on the self-bounding technique [Zimmert and Seldin, 2021]. Application of this technique in the no delay setting is relatively straightforward, but in presence of delays it requires control of the drift of the playing distribution from the moment an action is played to the moment the feedback arrives. Cesa-Bianchi et al. [2019] have bounded the drift of the playing distribution of the EXP3 algorithm in the uniform delays setting with a fixed learning rate. But best-of-both-worlds algorithms require decreasing learning rates [Mourtada and Gaïffas, 2019], which makes the drift control much more challenging. The problem gets even more challenging in the case of arbitrary delays, because it requires drift control over arbitrary long periods of time.

We apply an FTRL algorithm with the same hybrid regularizer as the one used by Zimmert and Seldin [2020], but with a different tuning of the learning rates. The new tuning has a minor effect on the adversarial regret bound, but allows us to make progress with the stochastic analysis. For the stochastic analysis we use the self-bounding technique. One of our key contributions is a general lemma that bounds the drift of the playing distribution derived from the time-varying hybrid regularizer over arbitrary delays. Using this lemma we derive near-optimal best-of-both-worlds regret guarantees for the case of fixed delays. But even with the lemma at hand, application of the self-bounding technique in presence of arbitrary delays is still much more challenging than in the no delays or fixed delay setting. Therefore, we resort to introducing an assumption of oracle knowledge of the maximal delay, which limits the maximal period of time over which we need to keep control over the drift. Our contributions are summarized below. To keep the presentation simple we assume uniqueness of the best arm throughout the paper. Tools for eliminating the uniqueness of the best arm assumption were proposed by Ito [2021].

1. We show that in the arbitrary delays setting with an oracle knowledge of the maximal delay $d_{max}$, our algorithm achieves $\mathcal{O}(\sqrt{KT}+\sqrt{D\log K}+d_{max}K^{1/3}\log K)$ regret bound in the adversarial regime simultaneously with $\mathcal{O}\left(\sum_{i\neq i^*}(\frac{\log T}{\Delta_i}+\frac{\sigma_{max}}{\Delta_i\log K})+d_{max}K^{1/3}\log K\right)$ regret bound in the stochastic regime, where $\sigma_{max}$ is the maximal number of outstanding observations. We note that $\sigma_{max}\leq d_{max}$, but it may potentially be much smaller. For example, if the first observation has a delay of $T$ and all the remaining observations have zero delay, then $d_{max}=T$, but $\sigma_{max}=1$.

2. In the case of uniform delays the above bounds simplify to $\mathcal{O}(\sqrt{KT}+\sqrt{dT\log K}+dK^{1/3}\log K)$ in the adversarial case and $\mathcal{O}\left(\sum_{i\neq i^*}(\frac{\log T}{\Delta_i}+\frac{d}{\Delta_i\log K})+dK^{1/3}\log K\right)$ in the stochastic case. For $T\geq dK^{2/3}\log K$ the last term in the adversarial regret bound is dominated by the middle term, which leads to the minimax optimal $\mathcal{O}(\sqrt{KT}+\sqrt{dT\log K})$ adversarial regret. The stochastic regret lower bound is trivially $\Omega(\min\{d\frac{\sum_{i\neq i^*}\Delta_i}{K},\sum_{i\neq i^*}\frac{\log T}{\Delta_i}\})=\Omega(d\frac{\sum_{i\neq i^*}\Delta_i}{K}+\sum_{i\neq i^*}\frac{\log T}{\Delta_i})$ and, therefore, our stochastic regret upper bound is near-optimal.

3. We present an $\Omega\left(\sqrt{KT}+\min_S(|S|+\sqrt{D_{\bar{S}}\log K})\right)$ regret lower bound for adversarial multi-armed bandits with non-uniformly delayed feedback, which matches the refined regret upper bound achieved by the skipping technique of Zimmert and Seldin [2020].

## 2  Problem setting

We study the multi-armed bandit with delays problem, in which at time $t=1,2,\ldots$ the learner chooses an arm $I_t$ among a set of $K$ arms and instantaneously suffers a loss $\ell_{t,I_t}$ from a loss vector $\ell_t\in[0,1]^K$ generated by the environment, but $\ell_{t,I_t}$ is not observed by the learner immediately. After a delay of $d_t$, at the end of round $t+d_t$, the learner observes the pair $(t,\ell_{t,I_t})$, namely, the loss and the index of the game round the loss is coming from. The sequence of delays $d_1,d_2,\ldots$ is selected arbitrarily by the environment. Without loss of generality we can assume that all the outstanding observations are revealed at the end of the game, i.e., $t+d_t\leq T$ for all $t$, where $T$ is the time horizon, unknown to the learner. We consider two regimes, oblivious adversarial and stochastic. The performance of the learner is evaluated using pseudo-regret, which is defined as

$$\overline{Reg}_T=\mathbb{E}\left[\sum_{t=1}^T\ell_{t,I_t}\right]-\min_{i\in[K]}\mathbb{E}\left[\sum_{t=1}^T\ell_{t,i}\right]=\mathbb{E}\left[\sum_{t=1}^T\left(\ell_{t,I_t}-\ell_{t,i_T^*}\right)\right],$$

where $i_T^*\in\operatorname{argmin}_{i\in[K]}\mathbb{E}\left[\sum_{t=t}^T\ell_{t,i}\right]$ is a best arm in hindsight in expectation over the loss generation model and the randomness of the learner. In the oblivious adversarial setting the losses are independent of the actions taken by the algorithm and considered to be deterministic, and the pseudo-regret is equal to the expected regret.

**Additional Notation:**  We use $\Delta^n$ to denote the probability simplex over $n+1$ points. The characteristic function of a closed convex set $\mathcal{A}$ is denoted by $\mathcal{I}_\mathcal{A}(x)$ and satisfies $\mathcal{I}_\mathcal{A}(x)=0$ for $x\in\mathcal{A}$ and $\mathcal{I}_\mathcal{A}(x)=\infty$ otherwise. The convex conjugate of a function $f:\mathbb{R}^n\to\mathbb{R}$ is defined by $f^*(y)=\sup_{x\in\mathbb{R}^n}\{\langle x,y\rangle-f(x)\}$. We also use bar to denote that the function domain is restricted to $\Delta^n$, e.g., $\bar{f}(x)=\begin{cases}f(x),&\text{if }x\in\Delta^n\\\infty,&\text{otherwise}\end{cases}$. We denote the indicator function of an event $\mathcal{E}$ by $\mathbb{1}(\mathcal{E})$ and use $\mathbb{1}_t(i)$ as a shorthand for $\mathbb{1}(I_t=i)$. The probability distribution over arms that is played by the learner at round $t$ is denoted by $x_t\in\Delta^{K-1}$.

## 3  Algorithm

The algorithm is based on Follow The Regularized Leader (FTRL) algorithm with the hybrid regularizer used by Zimmert and Seldin [2020], stated in equation (1). At each time step $t$ let $\sigma_t=\sum_{s=1}^{t-1}\mathbb{1}(s+d_s\geq t)$ be the number of outstanding observations and $\mathcal{D}_t=\sum_{s=1}^t\sigma_t$ be the

cumulative number of outstanding observations, then the learning rates are defined as

$$\eta_t^{-1} = \sqrt{t + \eta_0}, \qquad\qquad \gamma_t^{-1} = \sqrt{\frac{\sum_{s=1}^{t} \sigma_s + \gamma_0}{\log K}}, \qquad (2)$$

where $\eta_0 = 10 d_{max} + d_{max}^2 / \left( K^{1/3} \log(K) \right)^2$ and $\gamma_0 = 24^2 d_{max}^2 K^{2/3} \log(K)$. The update rule for the distribution over actions played by the learner is

$$x_t = \nabla \bar{F}_t^*(-\hat{L}_t^{obs}) = \arg\min_{x \in \Delta^{K-1}} \langle \hat{L}_t^{obs}, x \rangle + F_t(x), \qquad (3)$$

where $\hat{L}_t^{obs} = \sum_{s=1}^{t-1} \hat{\ell}_s \mathbb{1}(s + d_s < t)$ is the cumulative importance-weighted observed loss and $\hat{\ell}_s$ is an importance-weighted estimate of the loss vector $\ell_s$ defined by

$$\hat{\ell}_{t,i} = \frac{\ell_{t,i} \mathbb{1}(I_t = i)}{x_{t,i}}.$$

At the beginning of round $t$ the algorithm calculates the cumulative number of outstanding observations $\mathcal{D}_t$ and uses it to define the learning rate $\gamma_t$. Next, it uses the FTRL update rule defined in (3) to define a distribution over actions $x_t$ from which to draw action $I_t$. Finally, at the end of round $t$ it receives the delayed observations and updates the cumulative loss estimation vector accordingly, so that $\hat{L}_{t+1}^{obs} = \hat{L}_t^{obs} + \sum_{s=1}^{t} \hat{\ell}_s \mathbb{1}(s + d_s = t)$. The complete algorithm is provided in Algorithm 1.

---

**Algorithm 1:** FTRL with advance tuning for delayed bandit

---

1 **Initialize** $\mathcal{D}_0 = 0$ and $\hat{L}_1^{obs} = \mathbf{0}_K$ (where $\mathbf{0}_K$ is a zero vector in $\mathbb{R}^K$)
2 **for** $t = 1, \dots, n$ **do**
3      Set $\sigma_t = \sum_{s=1}^{t-1} \mathbb{1}(s + d_s > t)$
4      Update $\mathcal{D}_t = \mathcal{D}_{t-1} + \sigma_t$
5      Set $x_t = \arg\min_{x \in \Delta^{K-1}} \langle \hat{L}_t^{obs}, x \rangle + F_t(x)$          // $F_t$ is defined in (1) and $\eta_t$ and $\gamma_t$ in (2)
6      Sample $I_t \sim x_t$
7      Observe $(s, \ell_{s,I_s})$ for all $s$ that satisfy $s + d_s = t$
8      $\hat{L}_{t+1}^{obs} = \hat{L}_t^{obs} + \sum_{s=1}^{t} \hat{\ell}_s \mathbb{1}(s + d_s = t)$

---

## 4   Best-of-both-worlds regret bounds for Algorithm 1

In this section we provide best-of-both-worlds regret bounds for Algorithm 1. First, in Theorem 1 we provide regret bounds for an arbitrary delay setting, where we assume an oracle access to $d_{max}$. Then, in Corollary 2 we specialize the result to a fixed delay setting.

**Theorem 1.** *Assume that Algorithm 1 is given an oracle knowledge of $d_{max}$. Then its pseudo-regret for any sequence of delays and losses satisfies*

$$\overline{Reg}_T = \mathcal{O}(\sqrt{TK} + \sqrt{D \log K} + d_{max} K^{1/3} \log K).$$

*Furthermore, in the stochastic regime the pseudo-regret additionally satisfies*

$$\overline{Reg}_T = \mathcal{O}\left( \sum_{i \neq i^*} (\frac{1}{\Delta_i} \log(T) + \frac{\sigma_{max}}{\Delta_i \log K}) + d_{max} K^{1/3} \log K \right).$$

A sketch of the proof is provided in Section 5 and detailed constants are worked out in Appendix C. For fixed delays Theorem 1 gives the following corollary.

**Corollary 2.** *If the delays are fixed and equal to $d$, and $T \geq dK^{2/3} \log K$, then the pseudo-regret of Algorithm 1 always satisfies*

$$\overline{Reg}_T = \mathcal{O}(\sqrt{TK} + \sqrt{dT \log K})$$

*and in the stochastic setting it additionally satisfies*

$$\overline{Reg}_T = \mathcal{O}\left( \sum_{i \neq i^*} (\frac{1}{\Delta_i} \log(T) + \frac{d}{\Delta_i \log K}) + dK^{1/3} \log K \right).$$

In the adversarial regime with fixed delays $d$, regret lower bound is $\Omega\left(\sqrt{KT} + \sqrt{dT\log K}\right)$, whereas in the stochastic regime with fixed delays the regret lower bound is trivially $\Omega(d\frac{\sum_{i\neq i^*}\Delta_i}{K} + \sum_{i\neq i^*}\frac{\log T}{\Delta_i})$. Thus, in the adversarial regime the corollary yields the minimax optimal regret bound and in the stochastic regime it is near-optimal. More explicitly, it is optimal within a multiplicative factor of $\sum_{i\neq i^*}\frac{1}{\Delta_i\log K} + \frac{K^{4/3}\log K}{\sum_{i\neq i^*}\Delta_i}$ in front of $d$.

If we fix a total delay budget $D$, then uniform delays $d = D/T$ is a special case, and in this sense Theorem 1 is also optimal in the adversarial regime and near-optimal in the stochastic regime, although for non-uniform delays improved regret bounds can potentially be achieved by skipping. We also note that having the dependence on $\sigma_{max}$ in the middle term of the stochastic regret bound in Theorem 1 is better than having a dependence on $d_{max}$, since $\sigma_{max} \leq d_{max}$, and in some cases it can be significantly smaller, as shown in the example in the Introduction and quantified by the following lemma.

**Lemma 3.** *Let $d_{max}(S) = \max_{s\in S} d_s$, where $S \subseteq \{1,\ldots,T\}$ is a subset of rounds. Let $\bar{S} = \{1,\ldots,T\} \setminus S$ be the remaining rounds. Then*

$$\sigma_{max} \leq \min_{S\subseteq\{1,\ldots,T\}}\left\{|S| + d_{max}(\bar{S})\right\}.$$

A proof of Lemma 3 is provided in Appendix A.

Finally, we note that the result in Theorem 1 is easily extendable to the corrupted regime, because the proof relies on the same self-bounding technique as the one used by Zimmert and Seldin [2021]. If we denote by $B_T^{stoch}$ the regret upper bound in the stochastic regime in Theorem 1 and by $C$ the total corruption budget, then in the corrupted regime the regret would be $\mathcal{O}(B_T^{stoch} + \sqrt{B_T^{stoch}C})$. The proof is straightforward, following the lines of Zimmert and Seldin [2021], and, therefore, left out.

# 5  A proof sketch of Theorem 1

In this section we provide a sketch of a proof of Theorem 1. We provide a proof sketch for the stochastic bound in Section 5.1. Afterwards, in Section 5.2, we show how the analysis of Zimmert and Seldin [2020] gives the adversarial bound stated in Theorem 1.

## 5.1  Stochastic Bound

We start by providing a key lemma (Lemma 4) that controls the drift of the playing distribution derived from the time-varying hybrid regularizer over arbitrary delays. We then introduce a drifted version of the pseudo-regret defined in (4), for which we use the key lemma to show that the drifted version of the pseudo-regret is close to the actual one. As a result, it is sufficient to bound the drifted version. The analysis of the drifted pseudo-regret follows by the standard analysis of the FTRL algorithm [Lattimore and Szepesvári, 2020] that decomposes the pseudo-regret (drifted pseudo-regret in our case) into stability and penalty terms. Thereafter, we proceed by using Lemma 4 again, this time to bound the stability term in order to apply the self-bounding technique [Zimmert and Seldin, 2019], which yields logarithmic regret in the stochastic setting. Our key lemma is the following.

**Lemma 4** (The Key Lemma)**.** *For any $i \in [K]$ and $s, t \in [T]$, where $s \leq t$ and $t - s \leq d_{max}$, we have*

$$x_{t,i} \leq 2x_{s,i}.$$

A detailed proof of the lemma is provided in Appendix B. Below we explain the high level idea behind the proof.

*Proof sketch.* We know that $x_t = \nabla\bar{F}_t^*(-\hat{L}_t^{obs})$ and $x_s = \nabla\bar{F}_s^*(-\hat{L}_s^{obs})$, so we introduce $\tilde{x} = \nabla\bar{F}_s^*(-\hat{L}_t^{obs})$ as an auxiliary variable to bridge between $x_t$ and $x_s$. The analysis consists of two key steps and is based on induction on $(t, s)$.

**Deviation Induced by the Loss Shift:** This step controls the drift when we fix the learning rates and shift the cumulative loss. We prove the following inequality:

$$\tilde{x}_i \leq \frac{3}{2}x_{s,i}.$$

Note that this step uses the induction assumption for $(s, s - d_r)$ for all $r < s : r + d_r = s$.

**Deviation Induced by the Change of Regularizer:** In this step we bound the drift when the cumulative loss vector is fixed and we change the regularizer. We show that

$$x_{t,i} \leq \frac{4}{3} \tilde{x}_i.$$

Combining these two steps gives us the desired bound. A proof of these steps is provided in Appendix B. ■

We use Lemma 4 to relate the drifted pseudo-regret to the actual pseudo-regret. Let $A_t = \{s : s \leq t \text{ and } s + d_s = t\}$ be the set of rounds for which feedback arrives at round $t$. We define the observed loss vector at time $t$ as $\hat{\ell}_t^{obs} = \sum_{s \in A_t} \hat{\ell}_s$ and the drifted pseudo-regret as

$$\overline{Reg}_T^{drift} = \mathbb{E}\left[\sum_{t=1}^{T} \left(\langle x_t, \hat{\ell}_t^{obs} \rangle - \hat{\ell}_{t,i_T^*}^{obs}\right)\right]. \tag{4}$$

We rewrite the drifted regret as

$$\overline{Reg}_T^{drift} = \mathbb{E}\left[\sum_{t=1}^{T} \sum_{s \in A_t} \left(\langle x_t, \hat{\ell}_s \rangle - \hat{\ell}_{s,i_T^*}\right)\right]$$

$$= \sum_{t=1}^{T} \sum_{s \in A_t} \sum_{i=1}^{K} \mathbb{E}[x_{t,i}(\hat{\ell}_{s,i} - \hat{\ell}_{s,i_T^*})]$$

$$= \sum_{t=1}^{T} \sum_{s \in A_t} \sum_{i=1}^{K} \mathbb{E}[x_{t,i}]\Delta_i = \sum_{t=1}^{T} \sum_{i=1}^{K} \mathbb{E}[x_{t+d_t,i}]\Delta_i,$$

where when taking the expectation we use the facts that $\hat{\ell}_s$ has no impact on the determination of $x_t$ and that the loss estimators are unbiased. Using Lemma 4 we make a connection between pseudo-regret and the drifted version:

$$\overline{Reg}_T^{drift} = \sum_{t=1}^{T} \sum_{i=1}^{K} \mathbb{E}[x_{t+d_t,i}]\Delta_i \geq \sum_{t=1}^{T-d_{max}} \sum_{i=1}^{K} \frac{1}{2}\mathbb{E}[x_{t+d_{max},i}]\Delta_i$$

$$= \frac{1}{2} \sum_{t=d_{max}+1}^{T} \sum_{i=1}^{K} \mathbb{E}[x_{t,i}]\Delta_i$$

$$\geq \frac{1}{2} \sum_{t=1}^{T} \sum_{i=1}^{K} \mathbb{E}[x_{t,i}]\Delta_i - \frac{d_{max}}{2} = \frac{1}{2}\overline{Reg}_T - \frac{d_{max}}{2},$$

where the first inequality follows by Lemma 4, and the second inequality uses $\sum_{t=1}^{d_{max}} \mathbb{E}[x_{t,i}]\Delta_i \leq d_{max}$. As a result, we have $\overline{Reg}_T \leq 2\overline{Reg}_T^{drift} + d_{max}$ and it suffices to upper bound $\overline{Reg}_T^{drift}$. We follow the standard analysis of FTRL, which decomposes the drifted pseudo-regret into *stabiltiy* and *penalty* terms as

$$\overline{Reg}_T^{drift} = \mathbb{E}\left[\underbrace{\sum_{t=1}^{T} \langle x_t, \hat{\ell}_t^{obs} \rangle + \bar{F}_t^*(-\hat{L}_{t+1}^{obs}) - \bar{F}_t^*(-\hat{L}_t^{obs})}_{stability}\right] + \mathbb{E}\left[\underbrace{\sum_{t=1}^{T} \bar{F}_t^*(-\hat{L}_t^{obs}) - \bar{F}_t^*(-\hat{L}_{t+1}^{obs}) - \ell_{t,i_T^*}}_{penalty}\right].$$

For the penalty term we have the following bound by Abernethy et al. [2015]

$$penalty \leq \sum_{t=2}^{T} \left(F_{t-1}(x_t) - F_t(x_t)\right) + F_T(e_{i_T^*}) - F_1(x_1),$$

where $\mathrm{e}_{i_T^*}$ denotes a the unit vector in $\mathbb{R}^K$ with the $i_T^*$-th element being one and zero elsewhere. By replacing the closed form of the regularizer in this bound and using the facts that $\eta_t^{-1} - \eta_{t-1}^{-1} = \mathcal{O}(\eta_t)$, $\gamma_t^{-1} - \gamma_{t-1}^{-1} = \mathcal{O}(\sigma_t \gamma_t / \log K)$, and $x_{t,i_T^*}^{\frac{1}{2}} - 1 \le 0$, we obtain

$$penalty \le \mathcal{O}\left( \sum_{t=2}^{T} \sum_{i \ne i^*} \eta_t x_{t,i}^{\frac{1}{2}} + \sum_{t=2}^{T} \sum_{i=1}^{K} \frac{\sigma_t \gamma_t x_{t,i} \log(1/x_{t,i})}{\log K} \right) + 2\sqrt{\eta_0(K-1)} + \sqrt{\gamma_0 \log K}. \tag{5}$$

In order to control the stability term we derive Lemma 5.

**Lemma 5** (Stability). *Let $\upsilon_t = |A_t|$. For any $\alpha_t \le \gamma_t^{-1}$ we have*

$$stability \le \sum_{t=1}^{T} \sum_{i=1}^{K} 2 f_t^{''}(x_{t,i})^{-1} (\hat{\ell}_{t,i}^{obs} - \alpha_t)^2.$$

*Furthermore, $\alpha_t = \frac{\sum_{j=1}^{K} f^{''}(x_{t,j})^{-1} \hat{\ell}_{t,j}^{obs}}{\sum_{j=1}^{K} f^{''}(x_{t,j})^{-1}}$ satisfies $\alpha_t \le \gamma_t^{-1}$ and yields*

$$\mathbb{E}[stability] \le \sum_{t=1}^{T} \sum_{i \ne i^*} 2\gamma_t(\upsilon_t - 1)\upsilon_t \mathbb{E}[x_{t,i}]\Delta_i + \sum_{t=1}^{T} \sum_{s \in A_t} \sum_{i=1}^{K} 2\eta_t \mathbb{E}[x_{t,i}^{3/2} x_{s,i}^{-1}(1 - x_{s,i})]. \tag{6}$$

A proof of the stability lemma is provided in Appendix A.3. We apply Lemma 4 to (6) to give bounds $\upsilon_t x_{t,i} = \sum_{s \in A_t} x_{t,i} \le 2 \sum_{s \in A_t} x_{s,i}$ and $x_{t,i}^{3/2} x_{s,i}^{-1}(1 - x_{s,i}) \le 2^{3/2} x_{s,i}^{1/2}(1 - x_{s,i})$. Moreover, in order to remove the best arm $i^*$ from the summation in the later bound we use $x_{s,i^*}^{1/2}(1 - x_{s,i^*}) \le \sum_{i \ne i^*} x_{s,i} \le \sum_{i \ne i^*} x_{s,i}^{1/2}$. These bounds together with the facts that we can change the order of the summations and that each $t$ belongs to exactly one $A_s$, gives us the following stability bound

$$\mathbb{E}[stability] = \mathcal{O}\left( \sum_{t=1}^{T} \sum_{i \ne i^*} \eta_t \mathbb{E}[x_{t,i}^{1/2}] + \sum_{t=1}^{T} \sum_{i \ne i^*} \gamma_{t+d_t}(\upsilon_{t+d_t} - 1)\mathbb{E}[x_{t,i}]\Delta_i \right). \tag{7}$$

By combining (7), (5), and the fact that $\overline{Reg}_T \le 2\overline{Reg}_T^{drift} + d_{max}$, we show that there exist constants $a, b, c \ge 0$, such that

$$\overline{Reg}_T \le \mathbb{E}\left[ a \underbrace{\sum_{t=1}^{T} \sum_{i \ne i^*} \eta_t x_{t,i}^{1/2}}_{A} + b \underbrace{\sum_{t=1}^{T} \sum_{i \ne i^*} \gamma_{t+d_t}(\upsilon_{t+d_t} - 1)x_{t,i}\Delta_i}_{B} + c \underbrace{\sum_{t=2}^{T} \sum_{i=1}^{K} \frac{\sigma_t \gamma_t x_{t,i} \log(1/x_{t,i})}{\log K}}_{C} \right]$$
$$+ \underbrace{4\sqrt{\eta_0(K-1)} + 2\sqrt{\gamma_0 \log K} + d_{max}}_{D}. \tag{8}$$

**Self bounding analysis:** We use the self-bounding technique to write $\overline{Reg}_T = 4\overline{Reg}_T - 3\overline{Reg}_T$, and then based on (8) we have

$$\overline{Reg}_T \le \mathbb{E}\left[4aA - \overline{Reg}_T\right] + \mathbb{E}\left[4bB - \overline{Reg}_T\right] + \mathbb{E}\left[4cC - \overline{Reg}_T\right] + 4D. \tag{9}$$

For $D$ we can substitute the values of $\gamma_0$ and $\eta_0$ and get

$$D = \mathcal{O}(d_{max}(K-1)^{1/3} \log K). \tag{10}$$

Upper bounding $A$, $B$, and $C$ requires separate and elaborate analysis, which we do in Lemmas 6, 7 and 8, respectively. Proofs of these lemmas are provided in Appendix A.2.

**Lemma 6** (A bound for $4aA - \overline{Reg}_T$). *We have the following bound for any $a \ge 0$:*

$$4aA - \overline{Reg}_T \le \sum_{i \ne i^*} \frac{4a^2}{\Delta_i} \log(T/\eta_0 + 1) + 1. \tag{11}$$

Lemma 6 contributes the logarithmic (in $T$) term to the regret bound.

**Lemma 7** (A bound for $4bB - \overline{Reg}_T$). *Let $\upsilon_{max} = \max_{t \in [T]} \upsilon_t$, then for any $b \geq 0$:*

$$4bB - \overline{Reg}_T \leq 64b^2 \upsilon_{max} \log K. \tag{12}$$

It is evident that $\upsilon_{max} \leq \sigma_{max} \leq d_{max}$, so the bound in Lemma 7 contributes an $\mathcal{O}(d_{max} \log K)$ term to the regret bound.

**Lemma 8** (A bound for $4cC - \overline{Reg}_T$). *For any $c \geq 0$:*

$$4cC - \overline{Reg}_T \leq \sum_{i \neq i^*} \frac{128c^2 \sigma_{max}}{\Delta_i \log K}. \tag{13}$$

Part of the pseudo-regret bound that corresponds to Lemma 8 comes from the penalty term related to the negative entropy part of the regularizer. In this part, despite the fact that $\sigma_{max}$ can be much smaller than $d_{max}$ (Lemma 3), the $\sum_{i \neq i^*} \frac{\sigma_{max}}{\Delta_i \log K}$ term could be very large when the suboptimality gaps are small. In Appendix D we show how an asymmetric oracle learning rate $\gamma_{t,i} \simeq \gamma_t / \sqrt{\Delta_i}$ for the negative entropy regularizer can be used to remove the $\sum_{i \neq i^*} 1/\Delta_i$ factor in front of $\sigma_{max}$. The possibility of removing this factor without the oracle knowledge is left as an open question.

Finally, by plugging (10),(11),(12),(13) into (9) we obtain the desired regret bound.

## 5.2 Adversarial bound

For the adversarial regime we use the final bound of Zimmert and Seldin [2021], which holds for any non-increasing learning rates:

$$\overline{Reg}_T \leq \sum_{t=1}^{T} \eta_t \sqrt{K} + \sum_{t=1}^{T} \gamma_t \sigma_t + 2\eta_T^{-1} \sqrt{K} + \gamma_T^{-1} \log K.$$

It suffices to substitute the values of the learning rates and use Lemma 11 for function $\frac{1}{\sqrt{x}}$:

$$\overline{Reg}_T \leq \sum_{t=1}^{T} \frac{\sqrt{K}}{\sqrt{t + \eta_0}} + \sum_{t=1}^{T} \frac{\sigma_t \sqrt{\log K}}{\sqrt{D_t + \gamma_0}} + 2\sqrt{KT + K\eta_0} + \sqrt{\log(K)D_T + \gamma_0 \log(K)}$$

$$= \mathcal{O}\left( \sqrt{KT} + \sqrt{\log(K)D_T} + d_{max}K^{1/3} \log K \right).$$

# 6 Refined lower bound

In this section, we prove a tight lower bound for adversarial regret with arbitrary delays. Thune et al. [2019] have proposed a skipping technique to achieve refined regret upper bounds in the adversarial regime with non-uniform delays. The technique was improved by Zimmert and Seldin [2020], but it remained unknown whether the refined regret bounds for regimes with non-uniform delays are tight. We answer this question positively by showing that the regret bound of Zimmert and Seldin [2020] is not improvable without additional assumptions. We first derive a refined lower bound for full-information games with variable loss ranges, which might be of independent interest. A proof is provided in Appendix E.

**Theorem 9.** *Let $L_1 \geq L_2 \geq \cdots \geq L_T \geq 0$ be a non-increasing sequence of positive reals and assume that there exists a permutation $\rho : [T] \to [T]$, such that the losses at time $t$ are bounded in $[0, L_{\rho(t)}]^K$. The minimax regret $Reg^*$ in the corresponding adversarial full-information game satisfies*

$$Reg^* \geq \max \left\{ \frac{1}{2} \sum_{t=1}^{\lfloor \log_2(K) \rfloor} L_t, \frac{1}{32} \sqrt{\sum_{t=\lfloor \log_2(K) \rfloor}^{T} L_t^2 \log(K)} \right\}.$$

From here we can directly obtain a lower bound for the full-information game with variable delays. This implies the same lower bound for bandits, since we have strictly less information available.

**Corollary 10.** *Let $(d_t)_{t=1}^T$ be a sequence of non-increasing delays, such that $d_t \leq T + 1 - t$ and let an oblivious adversary select all loss vectors $(\ell_t)_{t=1}^T$ in $[0,1]^K$ before the start of the game. The minimax regret of the full-information game is bounded from below by*

$$Reg^* = \Omega\left(\min_{S \subset [T]} |S| + \sqrt{D_{\bar{S}} \log(K)}\right) , \ where \ D_{\bar{S}} = \sum_{t \in [T] \setminus S} d_t .$$

*Proof.* We divide the time horizon greedily into $M$ buckets, such that the actions for all timesteps inside a bucket have to be chosen before the first feedback from any timestep inside the bucket is received. In other words, let bucket $B_m = \{b_m, \ldots, b_{m+1} - 1\}$, then $\forall t \in B_m : t + d_t > b_{m+1} - 1$, while $\exists t \in B_m : t + d_t = b_{m+1}$. This division of buckets has the following properties:

(i) monotonically decreasing sizes: $|B_1| \geq |B_2| \geq \cdots \geq |B_M|$.

(ii) upper bound on the sum of delays: $\forall m \in [M-1] : |B_m|^2 \geq \sum_{t \in B_{m+1}} d_t$.

Both properties follow directly from the non-decreasing nature of the delays.

$$|B_m| = b_{m+1} - b_m \leq b_m + d_{b_m} - b_m = d_{b_m}$$
$$|B_m| = \min_{t \in B_m}\{d_t + t - b_m\} \geq d_{b_{m+1}-1} + \min_{t \in B_m}\{t - b_m\} \geq d_{b_{m+1}-1} .$$

Hence

$$|B_m| \geq d_{b_{m+1}-1} \geq d_{b_{m+1}} \geq |B_{m+1}| ,$$
$$\sum_{t \in B_{m+1}} d_t \leq |B_{m+1}| \cdot d_{b_{m+1}} \leq |B_{m+1}| \cdot |B_m| \leq |B_m|^2 .$$

Set $S' = \bigcup_{m=1}^{\lfloor \log_2(K) \rfloor} B_m$ and let the adversary set all losses within a bucket to the same value, then the game reduces to a full information game over $M$ rounds with loss ranges $|B_1|, |B_2|, \ldots, |B_M|$. Applying Theorem 9 yields

$$Reg^* \geq \max\left\{\frac{1}{2}\sum_{m=1}^{\lfloor \log_2(K) \rfloor} |B_m|, \frac{1}{32}\sqrt{\sum_{m=\lfloor \log_2(K) \rfloor}^{M} |B_m|^2 \log(K)}\right\}$$

$$\geq \max\left\{\frac{1}{2}|S'|, \frac{1}{32}\sqrt{\sum_{t \in \bar{S}'} d_t \log(K)}\right\} = \Omega\left(\min_{S \subset [T]} |S| + \sqrt{\sum_{t \in \bar{S}} d_t \log(K)}\right) .$$

∎

# 7 Discussion

We have presented a best-of-both-worlds analysis of a slightly modified version of the algorithm of Zimmert and Seldin [2020] for bandits with delayed feedback. The key novelty of our analysis is the control of the drift of the playing distribution over arbitrary, but bounded, time intervals when the learning rate is changing over time. This control is necessary for best-of-both-worlds guarantees, but it is much more challenging than the drift control over fixed time intervals with fixed learning rate that appeared in prior work.

We also presented an adversarial regret lower bound matching the skipping-based refined regret upper bound of Zimmert and Seldin [2020] within constants.

Our work leads to several exciting open questions. The main one is whether skipping can be used to eliminate the need in oracle knowledge of $d_{max}$. If possible, this would remedy the deterioration of the adversarial bound by the additive factor of $d_{max}$, because the skipping threshold would be dominated by $\sqrt{D_{\bar{S}} \log K}$. Another open question is whether the $\frac{\sigma_{max}}{\Delta_i}$ term can be eliminated from the stochastic bound. Yet another open question is whether the $d_{max}$ factor in the stochastic bound can be reduced to $\sigma_{max}$ and whether the multiplicative terms dependent on $K$ can be eliminated. An extension of the results to first order bounds, that depend on the cumulative loss of the best action rather than $T$, and extension to arm dependent delays are also open questions. For now it was only done in the adversarial setting [Gyorgy and Joulani, 2021, Van Der Hoeven and Cesa-Bianchi, 2022].

## Acknowledgments and Disclosure of Funding

This project has received funding from European Union's Horizon 2020 research and innovation programme under the Marie Skłodowska-Curie grant agreement No 801199. YS acknowledges partial support by the Independent Research Fund Denmark, grant number 9040-00361B.

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
