Hence

$$|B_m| \ge d_{b_{m+1}-1} \ge d_{b_{m+1}} \ge |B_{m+1}| ,$$
$$\sum_{t \in B_{m+1}} d_t \le |B_{m+1}| \cdot d_{b_{m+1}} \le |B_{m+1}| \cdot |B_m| \le |B_m|^2 .$$

Set $S' = \bigcup_{m=1}^{\lfloor \log_2(K) \rfloor} B_m$ and let the adversary set all losses within a bucket to the same value, then the game reduces to a full information game over $M$ rounds with loss ranges $|B_1|, |B_2|, \dots, |B_M|$. Applying Theorem 9 yields

$$Reg^* \ge \max \left\{ \frac{1}{2} \sum_{m=1}^{\lfloor \log_2(K) \rfloor} |B_m|, \frac{1}{32} \sqrt{\sum_{m=\lfloor \log_2(K) \rfloor}^{M} |B_m|^2 \log(K)} \right\}$$

$$\ge \max \left\{ \frac{1}{2} |S'|, \frac{1}{32} \sqrt{\sum_{t \in \bar{S}'} d_t \log(K)} \right\} = \Omega \left( \min_{S \subset [T]} |S| + \sqrt{\sum_{t \in \bar{S}} d_t \log(K)} \right) .$$
∎

# 7 Discussion

We have presented a best-of-both-worlds analysis of a slightly modified version of the algorithm of Zimmert and Seldin [2020] for bandits with delayed feedback. The key novelty of our analysis is the control of the drift of the playing distribution over arbitrary, but bounded, time intervals when the learning rate is changing over time. This control is necessary for best-of-both-worlds guarantees, but it is much more challenging than the drift control over fixed time intervals with fixed learning rate that appeared in prior work.

We also presented an adversarial regret lower bound matching the skipping-based refined regret upper bound of Zimmert and Seldin [2020] within constants.

Our work leads to several exciting open questions. The main one is whether skipping can be used to eliminate the need in oracle knowledge of $d_{max}$. If possible, this would remedy the deterioration of the adversarial bound by the additive factor of $d_{max}$, because the skipping threshold would be dominated by $\sqrt{D_{\bar{S}} \log K}$. Another open question is whether the $\frac{\sigma_{max}}{\Delta_i}$ term can be eliminated from the stochastic bound. Yet another open question is whether the $d_{max}$ factor in the stochastic bound can be reduced to $\sigma_{max}$ and whether the multiplicative terms dependent on $K$ can be eliminated. An extension of the results to first order bounds, that depend on the cumulative loss of the best action rather than $T$, and extension to arm dependent delays are also open questions. For now it was only done in the adversarial setting [Gyorgy and Joulani, 2021, Van Der Hoeven and Cesa-Bianchi, 2022].

## Acknowledgments and Disclosure of Funding

This project has received funding from European Union's Horizon 2020 research and innovation programme under the Marie Skłodowska-Curie grant agreement No 801199. YS acknowledges partial support by the Independent Research Fund Denmark, grant number 9040-00361B.

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

# A  Proofs of the lemmas for the analysis of Algorithm 1

## A.1  A proof of Lemma 3

*Proof.* Let $S \subseteq \{1, \ldots, T\}$ and $\bar{S} = \{1, \ldots, T\} \setminus S$ be an arbitrary split of the game rounds. Consider the number of outstanding observations $\sigma_t$ at an arbitrary round $t$. The number $\sigma_t$ is bounded by the sum of the number of outstanding observations from actions taken in the rounds in $S$ and the number of outstanding observations from actions taken in the rounds in $\bar{S}$. The former is bounded by $|S|$, and the latter is bounded by $d_{max}(\bar{S})$, since by definition of $d_{max}(\bar{S})$ any observation from an action taken in a round in $\bar{S}$ can be outstanding for at most $d_{max}(\bar{S})$ rounds. Since this holds for any split of the rounds $\{1, \ldots, T\}$ into $S$ and $\bar{S}$, we have $\sigma_{max} = \max_t \sigma_t \leq \min_{S \subseteq \{1, \ldots, T\}} \left( |S| + d_{max}(\bar{S}) \right)$. ∎

## A.2  Proofs of the lemmas supporting the proof of Theorem 1

We start with providing some auxiliary lemmas.

**Lemma 11** (Integral inequality: Lemma 4.13 of Orabona [2019]). *Let $g(x)$ be a positive nonincreasing function, then for any non-negative sequence $\{z_n\}_{n \in \{0, \ldots, N\}}$ we have*

$$\sum_{n=1}^{N} z_n g\left( \sum_{i=0}^{n} z_i \right) \leq \int_{z_0}^{\sum_{i=0}^{N} z_i} g(x) dx.$$

**Lemma 12.** *Let $\sigma_t$ and $\upsilon_t$ be the number of outstanding observations and arriving observations at time $t$, respectively, then the following inequality holds for all $t$*

$$\sum_{s=1}^{t} \sigma_s \geq \sum_{s=1}^{t} \frac{\upsilon_s^2 - \upsilon_s}{2}.$$

*Proof.* Note that $A_s = \{r : r + d_r = t\}$. We define $D_s = \{d_r : r \in A_s\}$ be the set of delays corresponding to observations that arrive at round $s$, then $D_s$ must have $\upsilon_s = |A_s|$ different number of elements, because $\forall r \in A_s : r + d_r = s$. As a result, we have

$$\sum_{r \in A_s} d_r \geq 0 + 1 + \ldots + (\upsilon_s - 1) = \frac{\upsilon_s(\upsilon_s - 1)}{2}.$$

This gives us the following inequality

$$\sum_{s=1}^{t} \frac{\upsilon_s^2 - \upsilon_s}{2} \leq \sum_{s=1}^{t} \sum_{r \in A_s} d_r$$
$$= \sum_{r : r + d_r \leq t} d_r.$$

On the other hand, $\sum_{s=1}^{t} \sigma_s \geq \sum_{r : r + d_r \leq t} d_r$, since every observation from an action taken at round $r$ with delay $d_r$ counts as outstanding over $d_r$ rounds, i.e., contributes 1 to $\sigma_{r+1}, \ldots, \sigma_{r+d_r}$, and observations that have not arrived by round $t$ contribute only to the left hand side of the inequality. Together with the preceding inequality this completes the proof. ∎

### A.2.1  A proof of Lemma 6

*Proof.* We bound $4aA - \overline{Reg}_T$.

$$4aA - \overline{Reg}_T = \sum_{t=1}^{T} \sum_{i \neq i^*} \left( \frac{4ax_{t,i}^{\frac{1}{2}}}{\sqrt{t + \eta_0}} - x_{t,i} \Delta_i \right)$$

$$\leq \sum_{t=1}^{T} \sum_{i \neq i^*} \frac{4a^2}{(t + \eta_0) \Delta_i} \leq \sum_{i \neq i^*} \frac{4a^2}{\Delta_i} \log(T/\eta_0 + 1) + 1, \tag{14}$$

where the first inequality uses the AM-GM inequality, by which for any $z$ and $y$ we have $z + y \geq 2\sqrt{zy} \Rightarrow 2\sqrt{zy} - y \leq z$. The second inequality follows by the integral bound on the harmonic series, by which $\sum_{t=1}^{T} 1/(t + \eta_0) \leq \log(T + \eta_0) - \log(\eta_0) + 1$. ∎

### A.2.2 Proof of Lemma 7

*Proof.* We have

$$4bB - \overline{Reg}_T = \sum_{t=1}^{T} \sum_{i \neq i^*} x_{t,i} \Delta_i \left( 4b(v_{t+d_t} - 1)\gamma_{t+d_t} - 1 \right).$$

We define $T_0$ to be the first round $t$ with $\gamma_t^{-1} \geq 4b(v_{max} - 1)$, where $v_{max} = \max_{s \in [T]}\{v_s\}$. Then in the summation over time, the rounds with $t + d_t \geq T_0$ provide a negative contribution, since $4b(v_{t+d_t} - 1)\gamma_{t+d_t} - 1 \leq \frac{4b(v_{t+d_t} - 1)}{4b(v_{max} - 1)} - 1 \leq 0$. Therefore,

$$4bB - \overline{Reg}_T \leq \sum_{t+d_t < T_0} \sum_{i \neq i^*} x_{t,i} \Delta_i \left( 4b(v_{t+d_t} - 1)\gamma_{t+d_t} - 1 \right)$$

$$\leq \sum_{t+d_t < T_0} 4b(v_{t+d_t} - 1)\gamma_{t+d_t} = \sum_{t=1}^{T_0-1} \sum_{s+d_s=t} 4b(v_t - 1)\gamma_t = \sum_{t=1}^{T_0-1} 4bv_t(v_t - 1)\gamma_t, \quad (15)$$

where the second inequality holds because $\sum_{i \neq i^*} x_{t,i} \Delta_i \leq 1$ and $v_{t+d_t} \geq 1$. For simplicity of notation, we denote $\tilde{v}_t = v_t(v_t - 1)/2$, for which Lemma 12 gives us $\sum_{s=1}^{t} \tilde{v}_t \leq \sum_{s=1}^{t} \sigma_s$. Therefore, we have

$$\sum_{t=1}^{T_0-1} 4bv_t(v_t - 1)\gamma_t \leq \sum_{t=1}^{T_0-1} \frac{8b\sqrt{\log K}\tilde{v}_t}{\sqrt{\sum_{s=1}^{t} \tilde{v}_t}}$$

$$\leq 16b\sqrt{(\log K) \sum_{t=1}^{T_0-1} \tilde{v}_t} \leq 16b\sqrt{(\log K) \sum_{t=1}^{T_0-1} \sigma_t} \leq 16b(\log K)\gamma_{T_0-1}^{-1}, \quad (16)$$

where the second inequality uses integral inequality Lemma 11 for $g(x) = \frac{1}{\sqrt{x}}$. Moreover, by the choice of $T_0$ we have $\gamma_{T_0-1}^{-1} \leq 4b(v_{max} - 1)$. Combining this with (15) and (16) gives us $4bB - \overline{Reg}_T \leq 64b^2 v_{max} \log K$. ∎

### A.2.3 Proof of Lemma 8

*Proof.* First, we remove $i^*$ from the summation in $C$ by using the following inequality

$$-x_{t,i^*} \log(x_{t,i^*}) \leq (1 - x_{t,i^*}) = \sum_{i \neq i^*} x_{t,i},$$

which follows by the fact that $z \log(z) + 1 - z$ is a decreasing function for $z \in [0, 1]$, and the minimum value is zero, therefore, it is non-negative for $z \in [0, 1]$. By using this inequality we have

$$\sum_{t=2}^{T} \sum_{i=1}^{K} \frac{-4c\sigma_t x_{t,i} \log(x_{t,i})}{\sqrt{(S_t + \gamma_0) \log K}} \leq 4c \underbrace{\sum_{t=1}^{T} \sum_{i \neq i^*} \frac{-\sigma_t x_{t,i} \log(x_{t,i})}{\sqrt{(S_t + \gamma_0) \log K}}}_{C_1} + 4c \underbrace{\sum_{t=1}^{T} \sum_{i \neq i^*} \frac{\sigma_t x_{t,i}}{\sqrt{(S_t + \gamma_0) \log K}}}_{C_2},$$

where $S_t = \sum_{s=1}^{t} \sigma_s$. We break the expression $4cC - \overline{Reg}_T$, into $4\left(cC_1 - \alpha\overline{Reg}_T\right) + 4\left(cC_2 - \beta\overline{Reg}_T\right)$, where $\alpha + \beta = 1/4$.

**Controlling $cC_2 - \beta\overline{Reg}_T$**

Let $\sigma_{max} = \max_{t \in [T]}\{\sigma_t\}$ and let $T_i$ be the first round $t$ when $S_t + \gamma_0 \geq \frac{c^2 \sigma_{max}^2}{\beta^2 \Delta_i^2 \log K}$. Then for all $t \geq T_i$ we have

$$\frac{c\sigma_t x_{t,i}}{\sqrt{(S_t + \gamma_0) \log K}} - \beta x_{t,i} \Delta_i \leq 0.$$

Therefore, rounds after $T_i$ provide negative contribution to the summation, and we have

$$cC_2 - \beta\overline{Reg}_T \leq \beta \sum_{i \neq i^*} \sum_{t=1}^{T_i-1} x_{t,i} \left( \frac{c\sigma_t}{\beta\sqrt{(S_t + \gamma_0)\log K}} - \Delta_i \right)$$

$$\leq \sum_{i \neq i^*} \sum_{t=1}^{T_i-1} \frac{c\sigma_t}{\sqrt{(S_t + \gamma_0)\log K}}$$

$$\leq \sum_{i \neq i^*} \frac{2c(\sqrt{S_{T_i-1} + \gamma_0} - \sqrt{\gamma_0})}{\sqrt{\log K}}$$

$$\leq \sum_{i \neq i^*} \frac{2c^2\sigma_{max}}{\beta\Delta_i\log K}, \tag{17}$$

where the third inequality uses Lemma 11 for $g(x) = \frac{1}{\sqrt{x}}$ and the last inequality follows by the choice of $T_i$, which gives $S_{T_i-1} + \gamma_0 \leq \frac{c^2\sigma_{max}^2}{\beta^2\Delta_i^2\log K}$.

**Controlling $cC_1 - \alpha\overline{Reg}_T$**

For $cC_1 - \alpha\overline{Reg}_T$, let $b_t = \frac{c\sigma_t}{\alpha\sqrt{(S_t+\gamma_0)\log K}}$, then

$$cC_1 - \alpha\overline{Reg}_T = \alpha\sum_{t=1}^{T}\sum_{i \neq i^*} (-b_t x_{t,i}\log(x_{t,i}) - \Delta_i x_{t,i})$$

$$\leq \alpha\sum_{t=1}^{T}\sum_{i \neq i^*} \max_{z \in [0,1]} \{-b_t z\log(z) - \Delta_i z\}.$$

The function $g(z) = -b_t z\log(z) - \Delta_i z$ is a concave function for $z \in [0,1]$ and the maximum occurs when the derivative is zero. So we must have $-b_t\log(z) - b_t - \Delta_i = 0 \Rightarrow z = e^{-\frac{\Delta_i}{b_t}-1}$, and by substitution $\max_{z \in [0,1]} g(z) = b_t e^{-\frac{\Delta_i}{b_t}-1}$. Therefore,

$$cC_1 - \alpha\overline{Reg}_T \leq \alpha\sum_{t=1}^{T}\sum_{i \neq i^*} b_t e^{-\frac{\Delta_i}{b_t}-1}$$

$$= \sum_{i \neq i^*}\sum_{t=1}^{T} \frac{c\sigma_t}{\sqrt{(S_t+\gamma_0)\log K}} \exp\left( -\frac{\alpha\Delta_i\sqrt{(S_t+\gamma_0)\log K}}{c\sigma_t} - 1 \right)$$

$$\leq \sum_{i \neq i^*}\sum_{t=1}^{T} \sigma_t \times \frac{c}{\sqrt{(S_t+\gamma_0)\log K}} \exp\left( -\frac{\alpha\Delta_i\sqrt{(S_t+\gamma_0)\log K}}{c\sigma_{max}} - 1 \right),$$

where $\sigma_{max} = \max_{t \in [T]}\{\sigma_t\}$. Let $g_i(x) = \frac{c}{\sqrt{x\log K}}\exp\left(-\frac{\alpha\Delta_i\sqrt{x\log K}}{c\sigma_{max}} - 1\right)$, then for each $i$ we need to upper bound $\sum_{t=1}^{T} \sigma_t g_i(S_t + \gamma_0)$, which by Lemma 11 can be upper bounded by $\int_{\gamma_0}^{S_T+\gamma_0} g_i(x)dx$, because $g$ is nonincreasing. On the other hand, for any $\delta, a \geq 0$, we have $\int \frac{a}{\sqrt{x}}\exp(-\frac{\delta\sqrt{x}}{a}-1)dx = -\frac{2a^2}{\delta}\exp(-\frac{\delta\sqrt{x}}{a}-1)$. So, using the closed form of $\int g_i(x)dx$ with $\delta = \frac{\alpha\Delta_i}{\sigma_{max}}, a = \frac{c}{\sqrt{\log K}}$, we have

$$cC_1 - \alpha\overline{Reg}_T \leq \sum_{i \neq i^*} \int_{\gamma_0}^{S_T+\gamma_0} g_i(x)dx$$

$$= \sum_{i \neq i^*} \frac{-2c^2\sigma_{max}}{\alpha\Delta_i\log K}\exp\left(-\frac{\alpha\Delta_i\sqrt{x\log K}}{c\sigma_{max}} - 1\right)\Bigg|_{x=\gamma_0}^{x=S_T+\gamma_0}$$

$$= \sum \frac{2c^2\sigma_{max}\left(\exp\left(-\frac{\alpha\Delta_i\sqrt{\gamma_0\log K}}{c\sigma_{max}} - 1\right) - \exp\left(-\frac{\alpha\Delta_i\sqrt{(S_T+\gamma_0)\log K}}{c\sigma_{max}} - 1\right)\right)}{\alpha\Delta_i\log K}$$

$$\leq \sum_{i \neq i^*} \frac{2c^2\sigma_{max}}{\alpha\Delta_i\log K}. \tag{18}$$

Taking together (17) and (18) gives us

$$4cC - \overline{Reg}_T \leq \sum_{i \neq i^*} \frac{8c^2 \sigma_{max}}{\Delta_i \log K} \left( \frac{1}{\beta} + \frac{1}{\alpha} \right) = \sum_{i \neq i^*} \frac{8c^2 \sigma_{max}}{\Delta_i \log K} \left( \frac{1}{1/4 - \alpha} + \frac{1}{\alpha} \right)$$

$$\leq \sum_{i \neq i^*} \frac{128c^2 \sigma_{max}}{\Delta_i \log K}, \tag{19}$$

where the second inequality uses $\alpha = \frac{1}{8}$.

■

## A.3 Proof of the stability lemma

The lemma has two parts, the first part is the general bound for the stability term and the second is a special case of that bound where we set $\alpha$ to a specific value to get the desirable bound.

Before starting the proof we provide one fact and one lemma that help us in the proof of the stability lemma. We recall that our regularization function is $F_t(x) = \sum_{i=1}^K f_t(x)$, where $f_t(x) = -2\eta_t^{-1}\sqrt{x} + \gamma_t^{-1}x(\log x - 1)$.

**Fact 13** ([Zimmert and Seldin, 2020]). $f_t^{*'}(x)$ *is a convex monotonically increasing function.*

*Proof.* The proof is available in Section 7.3 of the supplementary material of Zimmert and Seldin [2020]. ■

**Lemma 14.** *Let $D_F(x, y) = F(x) - F(y) - \langle x - y, \nabla F(y) \rangle$ be the Bergman divergence of a function $F$. Then for any $x \in \boldsymbol{dom}(f_t)$, and any $\ell$ such that $\ell \geq -\gamma_t^{-1}$:*

$$D_{f_t^*}(f_t^{'}(x) - \ell, f_t^{'}(x)) \leq \frac{\ell^2}{2f_t^{''}(ex)}.$$

*Moreover, it is easy to see $(f_t^{''}(ex))^{-1} \leq 4(f_t^{''}(x))^{-1}$, which implies $D_{f_t^*}(f_t^{'}(x) - \ell, f_t^{'}(x)) \leq \frac{2\ell^2}{f_t^{''}(x)}$.*

*Proof.* By Taylor's theorem there exists $\tilde{x} \in \left[ f_t^{*'}(f_t^{'}(x) - \ell), f_t^{*'}(f_t^{'}(x)) \right]$, such that

$$D_{f_t^*}(f_t^{'}(x) - \ell, f_t^{'}(x)) = \frac{1}{2}\ell^2 f_t^{*''}(f_t^{'}(\tilde{x})) = \frac{1}{2}\ell^2 f_t^{''}(\tilde{x})^{-1},$$

where the second equality is a property of the convex conjugate operation. We have two cases for $\ell$:

1. If $\ell \geq 0$, then based on Fact 13 we know that $f_t^{*'}$ is increasing, so $\tilde{x} \leq x$. On the other hand, $f^{''}(x)^{-1}$ is increasing, so $f_t^{''}(\tilde{x})^{-1} \leq f_t^{''}(x)^{-1} \leq f_t^{''}(ex)^{-1}$.

2. If $\ell < 0$, then $\tilde{x} \in \left[ f_t^{*'}(f_t^{'}(x)), f_t^{*'}(f_t^{'}(x) - \ell) \right]$. We show that $f_t^{*'}(f_t^{'}(x) - \ell) \leq ex$, which by the choice of $\tilde{x}$ implies $\tilde{x} \leq ex$, and consequently, like in the other case, we end up having $f_t^{''}(\tilde{x})^{-1} \leq f_t^{''}(ex)^{-1}$.

   Since $f^{*'}$ is increasing and $ex = f^{*'}(f^{'}(ex))$, it suffices to prove that $f^{'}(ex) \geq f^{'}(x) - \ell$, or, equivalently, $f^{'}(ex) - f^{'}(x) \geq -\ell$. So

   $$f^{'}(ex) - f^{'}(x) = \left( -\eta_t^{-1}(ex)^{-1/2} + \gamma_t^{-1}\log(ex) \right) - \left( -\eta_t^{-1}x^{-1/2} + \gamma_t^{-1}\log(x) \right)$$

   $$= \eta_t^{-1}x^{-1/2}\left( 1 - \frac{1}{\sqrt{2}} \right) + \gamma_t^{-1} \geq \gamma_t^{-1} \geq -\ell$$

■

***Proof of the First Part of the Stability Lemma.*** We have $x_t = \arg\min_{x \in \Delta^{K-1}} \langle \hat{L}_t^{obs}, x \rangle + F_t(x)$, so by the KKT conditions there exists $c_0 \in \mathbb{R}$, such that $-\hat{L}_t^{obs} = \nabla F_t(x_t) - c_0 \mathbf{1}_K$. On the other hand, $\bar{F}_t(-L + c\mathbf{1}_K) = \bar{F}_t(-L) + c$ for any $c \in \mathbb{R}$ and $L \in \mathbb{R}^K$ and the equality holds iff $c = 0$. Therefore, using these two facts we

can rewrite the stability term as

$$\sum_{t=1}^{T} \langle x_t, \hat{\ell}_t^{obs} \rangle + \bar{F}_t^*(-\hat{L}_{t+1}^{obs}) - \bar{F}_t^*(-\hat{L}_t^{obs}) = \sum_{t=1}^{T} \langle x_t, \hat{\ell}_t^{obs} - \alpha_t \mathbf{1}_K \rangle + \bar{F}_t^*(-\hat{L}_{t+1}^{obs} + (\alpha_t + c_0)\mathbf{1}_K) - \bar{F}_t^*(-\hat{L}_t^{obs} + c_0\mathbf{1}_K)$$

$$= \sum_{t=1}^{T} \langle x_t, \hat{\ell}_t^{obs} - \alpha_t \mathbf{1}_K \rangle + \bar{F}_t^*(\nabla F_t(x_t) - (\hat{\ell}_t^{obs} - \alpha_t \mathbf{1}_K)) - \bar{F}_t^*(\nabla F_t(x_t))$$

$$\leq \sum_{t=1}^{T} \langle x_t, \hat{\ell}_t^{obs} - \alpha_t \mathbf{1}_K \rangle + F_t^*(\nabla F_t(x_t) - (\hat{\ell}_t^{obs} - \alpha_t \mathbf{1}_K)) - F_t^*(\nabla F_t(x_t))$$

$$= \sum_{i=1}^{K} D_{f_t^*}\left( f_t'(x_{t,i}) - (\hat{\ell}_{t,i}^{obs} - \alpha_t), f_t'(x_{t,i}) \right), \qquad (20)$$

where the inequality holds because $\bar{F}_t^*(L) \leq F_t^*(L)$ for all $L \in \mathbb{R}^K$ and $\bar{F}_t^*(\nabla F_t(x)) = F_t^*(\nabla F_t(x))$ for all $x \in \mathbb{R}^K$. Hence, since $\alpha_t \leq \gamma_t^{-1}$, we have $\hat{\ell}_{t,i}^{obs} - \alpha_t \geq -\alpha_t \geq -\gamma_t^{-1}$. This implies that we can apply Lemma 14 to get the following bound for (20)

$$stability \leq \sum_{i=1}^{K} 2f_t''(x_{t,i})^{-1}(\hat{\ell}_{t,i}^{obs} - \alpha_t)^2.$$

∎

***Proof of the Second Part of the Stability Lemma.*** First, we must check whether $\alpha_t = \frac{\sum_{j=1}^{K} f''(x_{t,j})^{-1}\tilde{\ell}_{t,j}}{\sum_{j=1}^{K} f''(x_{t,j})^{-1}}$ satisfies $\alpha_t \leq \gamma_t^{-1}$ or not:

$$\alpha_t = \frac{\sum_{j=1}^{K} f''(x_{t,j})^{-1}\tilde{\ell}_{t,j}}{\sum_{j=1}^{K} f''(x_{t,j})^{-1}}$$

$$= \frac{\sum_{j=1}^{K} f''(x_{t,j})^{-1} \sum_{s \in A_t} \hat{\ell}_{s,j}}{\sum_{j=1}^{K} f''(x_{t,j})^{-1}}$$

$$\leq 8|A_t|(K-1)^{\frac{1}{3}} \leq 8d_{max}(K-1)^{\frac{1}{3}} \leq \gamma_t^{-1},$$

where the first inequality uses Lemma 17. To simplify the analysis, for all $i$ let $z_i = f_t''(x_{t,i})^{-1}$, then by substitution of the value of $\alpha_t$ in the stability expression we have

$$\sum_{i=1}^{K} z_i(\tilde{\ell}_{t,i} - \alpha_t)^2 = \sum_{i=1}^{K} z_i \tilde{\ell}_{t,i}^2 - 2\sum_{i=1}^{K} z_i \tilde{\ell}_{t,i}\alpha_t + \sum_{i=1}^{K} z_i \alpha_t^2$$

$$= \sum_{i=1}^{K} z_i \tilde{\ell}_{t,i}^2 - \frac{(\sum_{i=1}^{K} z_i \tilde{\ell}_{t,i})^2}{\sum_{i=1}^{K} z_i}$$

$$= \sum_{i=1}^{K} z_i \tilde{\ell}_{t,i}^2 - \frac{\sum_{i=1}^{K} z_i^2 \tilde{\ell}_{t,i}^2}{\sum_{i=1}^{K} z_i} - \frac{\sum_{i,j,i \neq j} z_i z_j \tilde{\ell}_{t,i}\tilde{\ell}_{t,j}}{\sum_{i=1}^{K} z_i}$$

$$= \sum_{i=1}^{K} \left( z_i - \frac{z_i^2}{\sum_{j=1}^{K} z_j} \right) \left( \sum_{s \in A_t} \hat{\ell}_{s,i} \right)^2 - \frac{\sum_{i,j,i \neq j} z_i z_j \left( \sum_{r,s \in A_t} \hat{\ell}_{r,i}\hat{\ell}_{s,j} \right)}{\sum_{i=1}^{K} z_i}$$

$$= \sum_{i=1}^{K} \left( z_i - \frac{z_i^2}{\sum_{j=1}^{K} z_j} \right) \left( \sum_{s \in A_t} \hat{\ell}_{s,i}^2 \right) \qquad (21)$$

$$+ \sum_{i=1}^{K} \left( z_i - \frac{z_i^2}{\sum_{j=1}^{K} z_j} \right) \left( \sum_{r,s \in A_t, r \neq s} \hat{\ell}_{r,i}\hat{\ell}_{s,i} \right) - \frac{\sum_{i,j,i \neq j} z_i z_j \left( \sum_{r,s \in A_t} \hat{\ell}_{s,i}\hat{\ell}_{r,j} \right)}{\sum_{i=1}^{K} z_i}.$$

$$(22)$$

We call the term in line (21) Stab1 and the two terms in line (22) Stab2. We first bound the expectation of Stab1.

$$\mathbb{E}[\text{Stab1}] = \mathbb{E}\left[\sum_{i=1}^{K}\left(z_i - \frac{z_i^2}{\sum_{i=1}^{K} z_i}\right)\left(\sum_{s\in A_t}\hat{\ell}_{s,i}^2\right)\right]$$

$$= \mathbb{E}\left[\sum_{i=1}^{K}\left(z_i - \frac{z_i^2}{\sum_{i=1}^{K} z_i}\right)\left(\sum_{s\in A_t}\mathbb{E}_s[\hat{\ell}_{s,i}^2]\right)\right]$$

$$= \mathbb{E}\left[\sum_{i=1}^{K}\left(z_i - \frac{z_i^2}{\sum_{i=1}^{K} z_i}\right)\left(\sum_{s\in A_t}\ell_{s,i}^2 x_{s,i}^{-1}\right)\right]$$

$$\leq \sum_{s\in A_t}\mathbb{E}\left[\sum_{i=1}^{K} z_i x_{s,i}^{-1} - \frac{\sum_{i=1}^{K} z_i^2 x_{s,i}^{-1}}{\sum_{i=1}^{K} z_i}\right]$$

$$\leq \sum_{s\in A_t}\mathbb{E}\left[\sum_{i=1}^{K} z_i x_{s,i}^{-1}(1 - x_{s,i})\right]$$

$$\leq \sum_{s\in A_t}\mathbb{E}\left[\sum_{i=1}^{K} 2\eta_t x_{t,i}^{3/2} x_{s,i}^{-1}(1 - x_{s,i})\right], \tag{23}$$

where the first inequality bounds losses by one and changes the order of summations, the second inequality uses Cauchy-Schwarz inequality $\sum_{i=1}^{K} z_i^2 x_{s,i}^{-1} = \left(\sum_{i=1}^{K} z_i^2 x_{s,i}^{-1}\right)\underbrace{\left(\sum_{i=1}^{K} x_{s,i}\right)}_{=1} \geq \left(\sum_{i=1}^{K} z_i\right)^2$, and the last

inequality uses the fact that $z_i = f_t''(x_{t,i})^{-1} \leq 2\eta_t x_{t,i}^{3/2}$.

For Stab2 we have

$$\mathbb{E}[\text{Stab2}] = \mathbb{E}\left[\frac{1}{\sum_{i=1}^{K} z_i}\left(\sum_{i=1}^{K}\sum_{r,s\in A_t, r\neq s}\sum_{j\neq i} z_i z_j \hat{\ell}_{r,i}\hat{\ell}_{s,i} - \sum_{i,j,i\neq j}\sum_{r,s\in A_t} z_i z_j \hat{\ell}_{s,i}\hat{\ell}_{r,j}\right)\right]$$

$$= \mathbb{E}\left[\frac{1}{\sum_{i=1}^{K} z_i}\left(\sum_{i=1}^{K}\sum_{r,s\in A_t, r\neq s}\sum_{j\neq i} z_i z_j \mu_i^2 - \sum_{i,j,i\neq j}\sum_{r,s\in A_t} z_i z_j \mu_i\mu_j\right)\right]$$

$$= \mathbb{E}\left[\frac{1}{\sum_{i=1}^{K} z_i}\left(\upsilon_t(\upsilon_t - 1)\sum_{i=1}^{K}\sum_{j\neq i} z_i z_j \mu_i^2 - \upsilon_t^2 \sum_{i,j,i\neq j} z_i z_j \mu_i\mu_j\right)\right] \tag{24}$$

$$\leq \mathbb{E}\left[\frac{\upsilon_t(\upsilon_t - 1)}{\sum_{i=1}^{K} z_i}\left(\sum_{i=1}^{K} z_i(\sum_{j=1}^{K} z_j)\mu_i^2 - \sum_{i=1}^{K} z_i^2\mu_i^2 - \sum_{i,j,i\neq j} z_i z_j \mu_i\mu_j\right)\right]$$

$$= \mathbb{E}\left[\frac{\upsilon_t(\upsilon_t - 1)}{\sum_{i=1}^{K} z_i}\left((\sum_{i=1}^{K} z_i\mu_i^2)(\sum_{i=1}^{K} z_i) - (\sum_{i=1}^{K} z_i\mu_i)^2\right)\right]$$

$$\leq \mathbb{E}\left[\frac{\upsilon_t(\upsilon_t - 1)}{\sum_{i=1}^{K} z_i}\left((\sum_{i=1}^{K} z_i\mu_i^2)(\sum_{i=1}^{K} z_i) - (\sum_{i=1}^{K} z_i)^2\mu_{i^*}^2\right)\right]$$

$$= \mathbb{E}\left[\upsilon_t(\upsilon_t - 1)(\sum_{i=1}^{K} z_i\mu_i^2 - \sum_{i=1}^{K} z_i\mu_{i^*}^2)\right]$$

$$\leq \mathbb{E}\left[\upsilon_t(\upsilon_t - 1)(\sum_{i\neq i^*} 2z_i\Delta_i)\right]$$

$$\leq \mathbb{E}\left[\sum_{i\neq i^*} 2\upsilon_t(\upsilon_t - 1)\gamma_t x_{t,i}\Delta_i\right], \tag{25}$$

where the second equality follows by the fact that for all $s \in A_t$, $x_s$ has no impact on $x_t$, and for all different elements of $A_t$, such as $r, s \in A_t$ and $r < s$, $x_r$ has no impact on $x_s$. Regarding the inequalities, the first one follows by $\upsilon_t^2 \geq \upsilon_t(\upsilon_t - 1)$, the second one holds because for all $i$ we have $\mu_i^* \leq \mu_i$, the third inequality follows by $\mu_i + \mu_{i^*} \leq 2$ and $\mu_i - \mu_{i^*} = \Delta_i$, and the last one substitutes $z_i = f''(x_{t,i})^{-1} \leq \gamma_t x_{t,i}$.

Combining (23) and (25) completes the proof. ■

## B  Proof of the Key Lemma

### B.1  Auxiliary results for the proof of the key lemma

First, we provide two facts and a lemma, which are needed for the proof of the key lemma. We recall that $f_t(x) = -2\eta_t^{-1}\sqrt{x} + \gamma_t^{-1}x(\log x - 1)$.

**Fact 15.** $f_t'(x)$ *is a concave function.*

*Proof.* $f_t'(x) = -\eta^{-1}x^{-1/2} + \gamma_t^{-1}\log x$, so the second derivative is $-\frac{3}{4}\eta^{-1}x^{-5/2} - \gamma_t^{-1}x^{-2} \le 0$. ■

**Fact 16.** $f_t''(x)^{-1}$ *is a convex function.*

*Proof.* Let $g(x) = f_t''(x)^{-1} = (\frac{\eta_t^{-1}x^{-3/2}}{2} + \gamma_t^{-1}x^{-1})^{-1}$, then the second derivative of $g(x)$ is

$$g''(x) = \frac{\eta_t\gamma_t^2 \cdot \left(2\eta_t x^{\frac{7}{2}} + 3\gamma_t x^3\right)}{2\sqrt{x}\left(2\eta_t x^{\frac{3}{2}} + \gamma_t x\right)^3},$$

which is positive. ■

**Lemma 17.** *Fix $t$ and $s$ where $t \ge s$, and assume that there exists $\alpha$, such that $x_{t,i} \le \alpha x_{s,i}$ for all $i \in [K]$, and let $f(x) = \left(-2\eta_t^{-1}\sqrt{x} + \gamma_t^{-1}x(\log x - 1)\right)$, then we have the following inequality*

$$\frac{\sum_{j=1}^K f''(x_{t,j})^{-1}\hat{\ell}_{s,j}}{\sum_{j=1}^K f''(x_{t,j})^{-1}} \le 2\alpha(K-1)^{\frac{1}{3}}.$$

*Proof for Lemma 17.* We begin the proof as the following

$$
\begin{aligned}
\frac{\sum_{i=1}^K f''(x_{t,i})^{-1}\hat{\ell}_{s,i}}{\sum_{i=1}^K f''(x_{t,i})^{-1}} &= \frac{f''(x_{t,i_s})^{-1}x_{s,i_s}^{-1}\ell_{s,i_s}}{\sum_{i=1}^K f''(x_{t,i})^{-1}} \\
&\le \frac{f''(x_{t,i_s})^{-1}x_{t,i_s}^{-1}(x_{t,i_s}/x_{s,i_s})}{\sum_{i=1}^K f''(x_{t,i})^{-1}} \\
&\le \frac{f''(x_{t,i_s})^{-1}\alpha x_{t,i_s}^{-1}}{\sum_{i=1}^K f''(x_{t,i})^{-1}} \\
&\le \frac{\alpha f''(x_{t,i_s})^{-1}x_{t,i_s}^{-1}}{(K-1)f''\left(\frac{1-x_{t,i_s}}{K-1}\right)^{-1} + f''(x_{t,i_s})^{-1}} \quad \text{Define } z := x_{t,i_s} \\
&= \frac{\alpha\left(\eta_t^{-1}z^{-3/2} + 2\gamma_t^{-1}z^{-1}\right)^{-1}z^{-1}}{(K-1)\left(\eta_t^{-1}(\frac{1-z}{K-1})^{-3/2} + 2\gamma_t^{-1}(\frac{1-z}{K-1})^{-1}\right)^{-1} + \left(\eta_t^{-1}z^{-3/2} + 2\gamma_t^{-1}z^{-1}\right)^{-1}} \\
&= \alpha\left((1-z)\frac{\eta_t^{-1}z^{-1/2} + 2\gamma_t^{-1}}{\eta_t^{-1}\sqrt{K-1}(1-z)^{-1/2} + 2\gamma_t^{-1}} + z\right)^{-1}, \quad (26)
\end{aligned}
$$

where the first inequality follows by $\ell_{s,i_s} \le 1$, the second one uses the assumption of the lemma that $x_{t,i} \le \alpha x_{s,i}$, and the third inequality is due to convexity of $f''(x)^{-1}$ from Fact 16. We consider two cases for $z$: $z < \frac{1}{K}$ and $z \ge \frac{1}{K}$.

  a) $z \le \frac{1}{K}$: This case implies

$$\frac{1-z}{z} = \frac{1}{z} - 1 \ge K - 1 \Rightarrow (1-z)^{-1/2}\sqrt{K-1} \le z^{-1/2}$$

$$\Rightarrow 1 \le \frac{\eta_t^{-1}z^{-1/2} + 2\gamma_t^{-1}}{\eta_t^{-1}\sqrt{K-1}(1-z)^{-1/2} + 2\gamma_t^{-1}}. \quad (27)$$

Plugging (27) into (26) gives

$$\frac{\sum_{i=1}^{K} f''(x_{t,i})^{-1}\hat{\ell}_{s,i}}{\sum_{i=1}^{K} f''(x_{t,i})^{-1}} \le \alpha\,(1 - z + z)^{-1} = \alpha.$$

b) $z \ge \frac{1}{K}$: Similar to the previous case, $z \ge \frac{1}{K}$ implies $\eta_t^{-1} z^{-1/2} \le \eta_t^{-1}\sqrt{K-1}(1-z)^{-1/2}$, so the minimum of $\frac{\eta_t^{-1} z^{-1/2} + 2\gamma_t^{-1}}{\eta_t^{-1}\sqrt{K-1}(1-z)^{-1/2} + 2\gamma_t^{-1}}$ occurs when $2\gamma_t^{-1} = 0$. Substitution of $2\gamma_t^{-1} = 0$ in (26) gives

$$\frac{\sum_{i=1}^{K} f''(x_{t,i})^{-1}\hat{\ell}_{s,i}}{\sum_{i=1}^{K} f''(x_{t,i})^{-1}} \le \alpha\left((1-z)^{3/2} z^{-1/2}(K-1)^{-1/2} + z\right)^{-1}. \tag{28}$$

Here we have the following two subcases

b1) $z \ge \frac{1}{(K-1)^{1/3}+1}$: This gives

$$\alpha\left((1-z)^{3/2} z^{-1/2}(K-1)^{-1/2} + z\right)^{-1} \le \alpha z^{-1} \le \alpha\left((K-1)^{1/3} + 1\right) \le 2\alpha(K-1)^{1/3}.$$

b2) $z \le \frac{1}{(K-1)^{1/3}+1}$: This implies $(1-z) \ge \frac{(K-1)^{1/3}}{(K-1)^{1/3}+1} \ge \frac{1}{2}$ and we can use it in (28) in the following way

$$\alpha\left((1-z)^{3/2} z^{-1/2}(K-1)^{-1/2} + z\right)^{-1} \le \alpha\left(\frac{z^{-1/2}(K-1)^{-1/2}}{\sqrt{8}} + z\right)^{-1}$$

$$= \alpha\left(\frac{z^{-1/2}(K-1)^{-1/2}}{2\sqrt{8}} + \frac{z^{-1/2}(K-1)^{-1/2}}{2\sqrt{8}} + z\right)^{-1}$$

$$\le \frac{\alpha}{3}\left(\frac{(K-1)^{-1}}{32}\right)^{-1/3} \le 2\alpha(K-1)^{1/3},$$

where the second inequality is by the AM-GM inequality.

Combining the results for all cases and setting $\alpha = 4$ we obtain the upper bound $8(K-1)^{1/3}$. ■

## B.2 Proof of the key lemma

*Proof of Lemma 4.* To show $x_{t,i} \le 2x_{s,i}$ for all $i$ we do induction on *valid* pairs $(t,s)$, where we call a pair $(t,s)$ valid if $s \le t$ and $t - s \le d_{max}$. The induction step for $(t,s)$ uses the induction assumption for all valid pairs $(t',s')$, such that $s',t' < t$, and all valid pairs $(t',s')$, such that $t' = t$ and $s < s' \le t$. Thus, the induction base would be all the pairs of $(t',t')$ for all $t' \in [T]$, for which the statement $x_{t',i} \le 2x_{t',i}$ trivially holds. Hence, it suffices to prove the induction step for the valid pair $(t,s)$.

As we mentioned in the proof sketch, we have $x_t = \bar{F}_t^*(-\hat{L}_t^{obs})$ and $x_s = \bar{F}_s^*(-\hat{L}_s^{obs})$, and we introduce $\tilde{x} = \bar{F}_s^*(-\hat{L}_t^{obs})$ as an auxiliary variable to bridge from $x_t$ and $x_s$. We bridge from $x_t$ to $x_s$ via $\tilde{x}$ in the following way.

**Deviation Induced by the Loss Shift:** This step controls the drift when we fix the regularization (more precisely, the learning rates) and shift the cumulative loss. We prove the following inequality:

$$\tilde{x}_i \le \frac{3}{2}x_{s,i}.$$

Note that this step uses the induction assumption for $(s, s - d_r)$ for all $r < s : r + d_r = s$.
**Deviation Induced by the Change of Regularizer:** In this step we bound the drift when the cumulative loss vector is fixed and we change the regularizer. We show that

$$x_{t,i} \le \frac{4}{3}\tilde{x}_i.$$

## Deviation induced by the change of regularizer

The regularizer at any round $r$ is $F_r(x) = \sum_{i=1}^{K} f_r(x_i) = \sum_{i=1}^{K}\left(-2\eta_r^{-1}\sqrt{x_i} + \gamma_r^{-1} x_i(\log x_i - 1)\right)$. Since $x_t = \nabla\bar{F}_t^*(-\hat{L}_t^{obs})$ and $\tilde{x} = \nabla\bar{F}_s^*(-\hat{L}_t^{obs})$, by the KKT conditions $\exists\mu, \tilde{\mu}$ s.t. $\forall i$:

$$f_s'(\tilde{x}_i) = -L_{s,i}^{obs} + \mu,$$

$$f_t'(x_{t,i}) = -L_{t,i}^{obs} + \tilde{\mu}.$$

We also know that $\exists j : \tilde{x}_j \geq x_{t,j}$ which leads to

$$-L_{t,j}^{obs} + \tilde{\mu} = f_t'(x_{t,j}) \leq f_s'(x_{t,j}) \leq f_s'(\tilde{x}_j) = -L_{s,j}^{obs} + \mu,$$

where the first inequality holds because the learning rates are decreasing, and the second inequality is due to the fact that $f_s'(x)$ is increasing. This implies that $\tilde{\mu} \leq \mu$, which gives us the following inequality for all $i$:

$$f_t'(x_{t,i}) = -\frac{1}{\eta_t\sqrt{x_{t,i}}} + \frac{\log(x_{t,i})}{\gamma_t} \leq -\frac{1}{\eta_s\sqrt{\tilde{x}_i}} + \frac{\log(\tilde{x}_i)}{\gamma_s} = f_s'(\tilde{x}_i).$$

Define $\alpha = x_{t,i}/\tilde{x}_i$. Using the above inequality we have

$$\frac{1}{\eta_s\sqrt{\tilde{x}_i}} - \frac{\log(\tilde{x}_i)}{\gamma_s} \leq \frac{1}{\eta_t\sqrt{\alpha\tilde{x}_i}} - \frac{\log(\tilde{x}_i)}{\gamma_t} - \frac{\log(\alpha)}{\gamma_t} \quad \text{(multiply both sides by } \eta_t\sqrt{\tilde{x}_i} \text{ and rearrange)}$$

$$\Rightarrow \frac{1}{\sqrt{\alpha}} \geq \frac{\eta_t}{\eta_s} + 2\sqrt{\tilde{x}_i}\log(\sqrt{\tilde{x}_i})\left(\frac{\eta_t}{\gamma_t} - \frac{\eta_t}{\gamma_s}\right) + \log(\alpha)\frac{\eta_t}{\gamma_t}\sqrt{\tilde{x}_i}$$

$$\geq \frac{\eta_t}{\eta_s} + \min_{0\leq z\leq 1}\left\{2z\log(z)\left(\frac{\eta_t}{\gamma_t} - \frac{\eta_t}{\gamma_s}\right) + \log(\alpha)\frac{\eta_t}{\gamma_t}z\right\}$$

$$\overset{(a)}{=} \frac{\eta_t}{\eta_s} - \frac{2}{e}\left(\frac{\eta_t}{\gamma_t} - \frac{\eta_t}{\gamma_s}\right)\left(\frac{1}{\sqrt{\alpha}}\right)^{\frac{\gamma_t^{-1}}{\gamma_t^{-1}-\gamma_s^{-1}}}$$

$$\overset{(b)}{\geq} \frac{\eta_t}{\eta_s} - \left(\frac{\eta_t}{\gamma_t} - \frac{\eta_t}{\gamma_s}\right)\frac{1}{\sqrt{\alpha}},$$

where (a) holds because the minimized function is convex and equating the first derivative to zero gives $z = \left(\frac{1}{\sqrt{\alpha}}\right)^{\frac{\gamma_t^{-1}}{\gamma_t^{-1}-\gamma_s^{-1}}}$, and (b) follows by $\frac{\gamma_t^{-1}}{\gamma_t^{-1}-\gamma_s^{-1}} \geq 1$ and $e \geq 2$. Rearranging the above result gives

$$\alpha \leq \left(\frac{\eta_s}{\gamma_t} - \frac{\eta_s}{\gamma_s} + \frac{\eta_s}{\eta_t}\right)^2 = \left(\eta_s(\gamma_t^{-1} - \gamma_s^{-1}) + \frac{\eta_s}{\eta_t}\right)^2. \tag{29}$$

Now we need to substitute the closed form of learning rates to obtain an upper bound for $\alpha$. As a reminder, the learning rates are

$$\gamma_s^{-1} = \frac{1}{\sqrt{\log K}}\sqrt{\sum_{r=1}^{s}\sigma_r + \gamma_0}, \quad \eta_s^{-1} = \sqrt{s + \eta_0},$$

$$\gamma_t^{-1} = \frac{1}{\sqrt{\log K}}\sqrt{\sum_{r=1}^{s+d}\sigma_r + \gamma_0}, \quad \eta_t^{-1} = \sqrt{s + d + \eta_0},$$

where $d = t - s$, $\eta_0 = 10d_{max} + d_{max}^2/\left(K^{1/3}\log(K)\right)^2$, and $\gamma_0 = 24^2 d_{max}^2 K^{2/3}\log(K)$. Therefore, in (29) we have

$$\eta_s\left(\gamma_t^{-1} - \gamma_s^{-1}\right) \leq \eta_s \frac{\sum_{r=s+1}^{s+d}\sigma_r}{\sqrt{\log(K)\left(\sum_{r=1}^{s+d}\sigma_r + \gamma_0\right)}}$$

$$\leq \eta_s \frac{\sum_{r=s+1}^{s+d}\sigma_r}{\sqrt{\log(K)\gamma_0}}$$

$$\leq \frac{d_{max}^2}{\sqrt{\log(K)\gamma_0\eta_0}} \leq \frac{d_{max}^2}{\sqrt{24^2 d_{max}^4}} = \frac{1}{24}, \tag{30}$$

where the third inequality follows by $d, \sigma_r \leq d_{max}$ for all $r$ and $\eta_s \leq \frac{1}{\sqrt{\eta_0}}$, and the last inequality holds because $\eta_0 \geq 16d_{max}^2/K^{2/3}$. On the other hand, for $\frac{\eta_s}{\eta_t}$ in (29) we have

$$\frac{\eta_s}{\eta_t} = \sqrt{\frac{s + d + \eta_0}{s + \eta_0}} = \sqrt{1 + \frac{d}{s + \eta_0}}$$

$$\leq \sqrt{1 + \frac{d}{10d_{max}}}$$

$$\leq \sqrt{1 + \frac{d_{max}}{10d_{max}}} = \sqrt{\frac{11}{10}}, \tag{31}$$

where the first and the second inequalities hold because $\eta_0 \geq 10 d_{max}$ and $d \leq d_{max}$, respectively. Plugging (30) and (31) into (29) gives us the following bound for $\alpha$:

$$\alpha \leq \left( \sqrt{\frac{11}{10}} + \frac{1}{24} \right)^2 \leq \frac{4}{3}. \tag{32}$$

**Deviation Induced by the Loss Shift**

We have $x_s = \nabla \bar{F}_s^*(-L_s^{obs})$ and $\tilde{x} = \nabla \bar{F}_s^*(-L_t^{obs})$. Since they both share the same regularizer $F_s(x) = \sum_{i=1}^K f_s(x_i)$, to simplify the notation we drop $s$ and use $f(x)$ to refer to $f_s(x)$. By the KKT conditions $\exists \mu, \tilde{\mu}$ s.t. $\forall i$:

$$f'(x_{s,i}) = -L_{s,i}^{obs} + \mu,$$
$$f'(\tilde{x}_i) = -L_{t,i}^{obs} + \tilde{\mu}.$$

Let $\tilde{\ell} = L_t^{obs} - L_s^{obs}$, then by the concavity of $f'(x)$ from Fact 15, we have

$$(x_{s,i} - \tilde{x}_i) f''(x_{s,i}) \leq \underbrace{f'(x_{s,i}) - f'(\tilde{x}_i)}_{\mu - \tilde{\mu} + \tilde{\ell}_i} \leq (x_{s,i} - \tilde{x}_i) f''(\tilde{x}_i). \tag{33}$$

Since $f''(x_{s,i}) \geq 0$, from the left side of (33) we get $x_{s,i} - \tilde{x}_i \leq f''(x_{s,i})^{-1} \left( \mu - \tilde{\mu} + \tilde{\ell}_i \right)$. Taking summation over all $i$ and using the fact that both vectors $x_s$ and $\tilde{x}$ are probability vectors, we have

$$0 = \sum_{i=1}^K x_{s,i} - \tilde{x}_i \leq \sum_{i=1}^K f''(x_{s,i})^{-1} \left( \mu - \tilde{\mu} + \tilde{\ell}_i \right)$$

$$\Rightarrow \tilde{\mu} - \mu \leq \frac{\sum_{i=1}^K f''(x_{s,i})^{-1} \tilde{\ell}_i}{\sum_{i=1}^K f''(x_{s,i})^{-1}}. \tag{34}$$

Combining the right hand sides of (33) and (34) gives

$$(\tilde{x}_i - x_{s,i}) f''(\tilde{x}_i) \leq \tilde{\mu} - \mu - \tilde{\ell}_i \leq \frac{\sum_{j=1}^K f''(x_{s,j})^{-1} \tilde{\ell}_j}{\sum_{j=1}^K f''(x_{s,j})^{-1}}$$

and by rearrangement

$$\tilde{x}_i \leq x_{s,i} + f''(\tilde{x}_i)^{-1} \times \frac{\sum_{j=1}^K f''(x_{s,j})^{-1} \tilde{\ell}_j}{\sum_{j=1}^K f''(x_{s,j})^{-1}}$$

$$\leq x_{s,i} + \gamma_s \tilde{x}_i \times \frac{\sum_{j=1}^K f''(x_{s,j})^{-1} \tilde{\ell}_j}{\sum_{j=1}^K f''(x_{s,j})^{-1}}, \tag{35}$$

where the last inequality holds because $f''(\tilde{x}_i)^{-1} = \left( \eta_s^{-1} \frac{1}{2} \tilde{x}_i^{-3/2} + \gamma_s^{-1} \tilde{x}_i^{-1} \right)^{-1}$. The next step for bounding $\tilde{x}_i$ is to bound $\frac{\sum_{j=1}^K f''(x_{s,j})^{-1} \tilde{\ell}_j}{\sum_{j=1}^K f''(x_{s,j})^{-1}}$ in (35), where $\tilde{\ell}_j = \sum_{r \in A} \hat{\ell}_{r,j}$ and $A = \{r : s \leq r + d_r < t\}$.

If there exists $r \in A$, such that $r > s$ and $2x_{r,i} \leq x_{s,i}$, then combining it with the induction assumption for $(t, r)$, i.e., $x_{t,i} \leq 2x_{r,i}$, leads to $x_{t,i} \leq 2x_{r,i} \leq x_{s,i}$, which completes the proof. Otherwise, that for all $r \in A$ we have either $r \leq s$ or $x_{s,i} \leq 2x_{r,i}$. If $r \leq s$, we can use the induction assumption for $(s, r)$, which gives $x_{s,i} \leq 2x_{r,i}$. Consequently, in either case, the inequality $x_{s,i} \leq 2x_{r,i}$ holds for all $r \in A$, and we can plug it into Lemma 17 to get the following bound for all $r \in A$:

$$\frac{\sum_{j=1}^K f''(x_{s,j})^{-1} \hat{\ell}_{r,j}}{\sum_{j=1}^K f''(x_{s,j})^{-1}} \leq 4(K-1)^{\frac{1}{3}}. \tag{36}$$

We then proceed by doing a summation over all $r \in A$ on both sides of the above inequality and get $\frac{\sum_{j=1}^{K} f''(x_{s,j})^{-1} \tilde{\ell}_j}{\sum_{j=1}^{K} f''(x_{s,j})^{-1}} \le 4|A|(K-1)^{\frac{1}{3}}$. Now it suffices to plug this result into (35):

$$\tilde{x}_i \le x_{s,i} + 4|A|\gamma_s \tilde{x}_i (K-1)^{\frac{1}{3}} \Rightarrow$$

$$\tilde{x}_i \le x_{s,i} \times \left( \frac{1}{1 - 4|A|\gamma_s(K-1)^{1/3}} \right) \tag{37}$$

$$\le x_{s,i} \times \left( \frac{1}{1 - 8\gamma_s d_{max}(K-1)^{1/3}} \right)$$

$$\le x_{s,i} \times \left( \frac{1}{1 - 8\sqrt{\log K/\gamma_0} d_{max}(K-1)^{1/3}} \right) = \frac{x_{s,i}}{1 - 1/3} = \frac{3}{2} x_{s,i}, \tag{38}$$

where the third inequality uses $|A| \le d_{max} + t - s \le 2d_{max}$, and the last one uses the facts that $\gamma_s \le \sqrt{\log(K)/\gamma_0}$ and $\gamma_0 = 24^2 d_{max}^2 (K-1)^{2/3} \log(K)$.

Combining (38) and (32) completes the proof. ∎

## C   Detailed constant factors in the regret bound for Algorithm 1

In this section we provide a detailed regret bound for Algorithm 1.
As we proved in Section 5 we have the following inequality for the drifted regret:

$$\overline{Reg}_T \le 2\overline{Reg}_T^{drift} + d_{max} \tag{39}$$

We first derive a bound for the drifted regret by splitting the drifted regret into stability and penalty terms, as mentioned in Section 5. Following the general analysis of the penalty term for FTRL [Abernethy et al., 2015], we have

$$penalty \le \sum_{t=2}^{T} (F_{t-1}(x_t) - F_t(x_t)) + F_T(x^*) - F_1(x_1),$$

which gives us

$$penalty = \sum_{t=2}^{T} \left( 2(\sum_{i=1}^{K} x_{t,i}^{\frac{1}{2}} - 1)(\eta_t^{-1} - \eta_{t-1}^{-1}) - \sum_{i=1}^{K} x_{t,i} \log(x_{t,i})(\gamma_t^{-1} - \gamma_{t-1}^{-1}) \right) - 2\eta_1^{-1} + 2\sqrt{K}\eta_1^{-1} + \gamma_1^{-1} \log K$$

$$\le \sum_{t=2}^{T} \left( 2\sum_{i \ne i^*} x_{t,i}^{\frac{1}{2}} (\eta_t^{-1} - \eta_{t-1}^{-1}) - \sum_{i=1}^{K} x_{t,i} \log(x_{t,i})(\gamma_t^{-1} - \gamma_{t-1}^{-1}) \right) + 2\sqrt{\eta_0(K-1)} + \sqrt{\gamma_0 \log K}$$

$$\le \sum_{t=2}^{T} \left( 2\sum_{i \ne i^*} \eta_t x_{t,i}^{\frac{1}{2}} - \sum_{i=1}^{K} \frac{\sigma_t \gamma_t x_{t,i} \log(x_{t,i})}{\sqrt{\log K}} \right) + 2\sqrt{\eta_0(K-1)} + \sqrt{\gamma_0 \log K}, \tag{40}$$

where the first inequality holds because $x_{t,i^*}^{\frac{1}{2}} \le 1$ and the second inequality follows by $\eta_t^{-1} - \eta_{t-1}^{-1} = \sqrt{t + \eta_0} - \sqrt{t - 1 + \eta_0} \le \frac{1}{\sqrt{t + \eta_0}} = \eta_t$ and $\gamma_t^{-1} - \gamma_{t-1}^{-1} = \frac{\gamma_t^{-2} - \gamma_{t-1}^{-2}}{\gamma_t^{-1} + \gamma_{t-1}^{-1}} \le \frac{\gamma_t^{-2} - \gamma_{t-1}^{-2}}{\gamma_t^{-1}}$.

For the stability term, we start from the bound given by Lemma 5:

$$\mathbb{E}[stability] \le \sum_{t=1}^{T} \sum_{i \ne i^*} 2\gamma_t(\upsilon_t - 1)\upsilon_t \mathbb{E}[x_{t,i}]\Delta_i + \sum_{t=1}^{T} \sum_{s \in A_t} \sum_{i=1}^{K} \eta_t \mathbb{E}[x_{t,i}^{3/2} x_{s,i}^{-1}(1 - x_{s,i})]. \tag{41}$$

In above inequality, we know that $\upsilon_t x_{t,i} = \sum_{s \in A_t} x_{t,i}$, and by Lemma 4 we have $x_{t,i} \le 2x_{s,i}$ for $s \in A_t$. Then for the first term in (41):

$$\sum_{t=1}^{T} \sum_{i \ne i^*} 2\gamma_t(\upsilon_t - 1)\upsilon_t x_{t,i}\Delta_i \le \sum_{t=1}^{T} \sum_{i \ne i^*} \sum_{s \in A_t} 4\gamma_t(\upsilon_t - 1)\upsilon_t x_{s,i}\Delta_i = \sum_{t=1}^{T} \sum_{i \ne i^*} 4\gamma_{t+d_t}(\upsilon_{t+d_t} - 1)x_{t,i}\Delta_i. \tag{42}$$

Furthermore, we can bound $x_{t,i}^{3/2} x_{s,i}^{-1}(1 - x_{s,i}) \le 2^{3/2} x_{s,i}^{1/2}(1 - x_{s,i})$. Moreover, in order to remove the best arm $i^*$ from the summation in the later bound we use $x_{t,i^*}^{3/2} x_{s,i^*}^{-1}(1 - x_{s,i^*}) \le 2\sum_{i \ne i^*} x_{s,i} \le \sum_{i \ne i^*} 2x_{s,i}^{1/2}$.

For the second term in (41) we have

$$\sum_{t=1}^{T}\sum_{s\in A_t}\sum_{i=1}^{K}\eta_t x_{t,i}^{3/2}x_{s,i}^{-1}(1-x_{s,i}) \leq \sum_{t=1}^{T}\sum_{s\in A_t}\sum_{i=1}^{K}\eta_t 2^{3/2}x_{s,i}^{1/2}(1-x_{s,i})$$

$$\leq \sum_{t=1}^{T}\sum_{s\in A_t}\sum_{i\neq i^*}\sqrt{8}\eta_t x_{s,i}^{1/2} + \sum_{t=1}^{T}\sum_{s\in A_t}\sum_{i\neq i^*}2\eta_t x_{s,i}^{1/2}$$

$$\leq \sum_{t=1}^{T}\sum_{i\neq i^*}5\eta_t x_{t,i}^{1/2}, \tag{43}$$

where the last inequality follows by the facts that we can change the order of the summations and that each $t$ belongs to exactly one $A_s$. Plugging (42) and (43) into (41) we have

$$\mathbb{E}[stability] \leq \mathbb{E}\left[\sum_{t=1}^{T}\sum_{i\neq i^*}4\gamma_{t+d_t}(\upsilon_{t+d_t}-1)x_{t,i}\Delta_i + \sum_{t=1}^{T}\sum_{i\neq i^*}5\eta_t x_{t,i}^{1/2}\right]. \tag{44}$$

Now it suffices to combine (44), (40), and (39) to get

$$\overline{Reg}_T \leq \mathbb{E}\left[14\underbrace{\sum_{t=1}^{T}\sum_{i\neq i^*}\eta_t x_{t,i}^{1/2}}_{A} + 8\underbrace{\sum_{t=1}^{T}\sum_{i\neq i^*}\gamma_{t+d_t}(\upsilon_{t+d_t}-1)x_{t,i}\Delta_i}_{B} + 2\underbrace{\sum_{t=2}^{T}\sum_{i=1}^{K}\frac{\sigma_t\gamma_t x_{t,i}\log(1/x_{t,i})}{\log K}}_{C}\right]$$

$$+ \underbrace{4\sqrt{\eta_0(K-1)} + 2\sqrt{\gamma_0\log K} + d_{max}}_{D}. \tag{45}$$

We rewrite the regret as

$$\overline{Reg}_T = 4\overline{Reg}_T - 3\overline{Reg}_T \leq 4\times 14A - \overline{Reg}_T + 4\times 8B - \overline{Reg}_T + 4\times 2C - \overline{Reg}_T + 4D,$$

where by applying Lemmas 6, 7, and 8 we achieve

$$4\times 14A - \overline{Reg}_T \leq \sum_{i\neq i^*}\frac{28^2}{\Delta_i}\log(T/\eta_0+1)$$

$$4\times 8B - \overline{Reg}_T \leq 64^2\upsilon_{max}\log K$$

$$4\times 2C - \overline{Reg}_T \leq \sum_{i\neq i^*}\frac{512\sigma_{max}}{\Delta_i\log K}.$$

Therefore, the final regret bound is

$$\overline{Reg}_T \leq \sum_{i\neq i^*}\frac{28^2}{\Delta_i}\log(T/\eta_0+1) + 64^2\upsilon_{max}\log K + \sum_{i\neq i^*}\frac{512\sigma_{max}}{\Delta_i\log K}$$

$$+ 16\sqrt{\eta_0(K-1)} + 8\sqrt{\gamma_0\log K} + 4d_{max}.$$

## D  Removing the multiplicative factor $1/\Delta_i$ from $\sigma_{max}/\Delta_i$ in the regret bound

In this section we discuss how an asymmetric *oracle* learning rate $\gamma_{t,i} \simeq \gamma_t/\sqrt{\Delta_i}$ for negative entropy regularizer can be used to remove the factor $\sum_{i\neq i^*}1/\Delta_i$ in front of $\sigma_{max}$ in the regret bound.

In the analysis of Algorithm 1 we divided the regret into stability and penalty expressions. Moreover, in each of the bounds for stability and penalty we have two terms which correspond to negative entropy and Tsallis parts of the hybrid regularizer. The terms related to negative entropy part in both stability and penalty bounds are

$$\underbrace{\sum_{t=1}^{T}\sum_{i\neq i^*}\gamma_{t+d_t}(\upsilon_{t+d_t}-1)\mathbb{E}[x_{t,i}]\Delta_i}_{B} + \underbrace{\sum_{i=1}^{K}\mathbb{E}[x_{t,i}\log(1/x_{t,i})](\gamma_t^{-1}-\gamma_{t-1}^{-1})}_{C},$$

where $B$ and $C$, as we have seen in Section 5, are due to stability and penalty terms, respectively. The idea here is to scale-up $\gamma_t$ to decrease $C$, however increasing $\gamma_t$ increases $B$. Hence, we are facing a trade off here. To deal with this trade-off we change the learning rates for negative entropy from symmetric $\gamma_t$ to asymmetric

$\gamma_{t,i}$, and we expect this change only affect the parts of regret bound come from the negative entropy part of the regularizer, which are $B$ and $C$. This change results in to having two following terms instead,

$$\underbrace{\sum_{t=1}^{T} \sum_{i \neq i^*} \gamma_{t+d_t,i}(v_{t+d_t} - 1)\mathbb{E}[x_{t,i}]\Delta_i}_{B_{new}} + \underbrace{\sum_{i=1}^{K} \mathbb{E}[x_{t,i}\log(1/x_{t,i})](\gamma_{t,i}^{-1} - \gamma_{t-1,i}^{-1})}_{C_{new}}.$$

Here if we could choose $\gamma_{t,i} = \gamma_t/\sqrt{\Delta_i}$, then using the definition of $\gamma_t$ we would be able to rewrite $B_{new}$ and $C_{new}$ as

$$B_{new} = \mathcal{O}\left(\sum_{t=1}^{T} \sum_{i \neq i^*} \gamma_{t+d_t}(v_{t+d_t} - 1)\mathbb{E}[x_{t,i}]\sqrt{\Delta_i}\right)$$

$$C_{new} = \mathcal{O}\left(\sum_{i=1}^{K} \frac{\sigma_t \gamma_t \mathbb{E}[x_{t,i}\log(1/x_{t,i})]\sqrt{\Delta_i}}{\sqrt{\log K}}\right).$$

Now we must see what is the result of applying the self-bounding technique on these new terms. For $B_{new}$ and $C_{new}$, following the similar analysis as Lemma 7 and Lemma 8 we can get

$$4B_{new} - \overline{Reg}_T = \mathcal{O}(v_{max}\log K) = \mathcal{O}(d_{max}\log K)$$

$$4C_{new} - \overline{Reg}_T = \mathcal{O}(\frac{\sigma_{max}}{\log K}).$$

This implies that injecting $\sqrt{1/\Delta_i}$ in the negative entropy learning rates removes the factor $\sum_{i \neq i^*} \frac{1}{\Delta_i}$ in front of the $\sigma_{max}$. More interestingly this comes without having any significant changes in the other terms of regret bound.

As a result, we conjecture that replacing a good estimation of the suboptimal gaps namely $\hat{\Delta}_i$ in $\gamma_{t,i}$ as $\gamma_{t,i} = \gamma_t/\sqrt{\hat{\Delta}_i}$ might be also helpful to remove the multiplicative factors related to suboptimal gaps in front of the $\sigma_{max}$. We leave this problem to future work.

# E   Lower bounds

---
**Algorithm 2:** Adversarial choice of $\ell$

---
**Input:** $x$
1 **Initialize** $\mathcal{I} = \{\arg\max_i x_i\}$ **while** $\sum_{i \in \mathcal{I}} x_i + \min_{i \in \bar{\mathcal{I}}} x_i \leq \frac{2}{3}$ **do**
2   $\quad\lfloor$ Update $\mathcal{I} \leftarrow \mathcal{I} \cup \{\arg\min_{i \in \bar{\mathcal{I}}} x_i\}$

3 **return** $\ell_i = \begin{cases} \min\{1, \frac{\sum_{i \in \bar{\mathcal{I}}} x_i}{\sum_{i \in \mathcal{I}} x_i}\} & \text{for } i \in \mathcal{I} \\ \max\{-1, -\frac{\sum_{i \in \mathcal{I}} x_i}{\sum_{i \in \bar{\mathcal{I}}} x_i}\} & \text{for } i \in \bar{\mathcal{I}} \end{cases}$

---

**Lemma 18.** *For any $x \in \Delta([K])$, such that $\max_i x_i \leq \frac{2}{3}$, the vector $\ell$ returned by Algorithm 2 satisfies $\ell \in [-1, 1]$, $\langle x, \ell \rangle = 0$, and $\sum_{i=1}^{K} x_i \ell_i^2 \geq \frac{1}{2}$.*

*Proof.* The first two properties follow directly by construction. For the third property we bound the ratio of the two sets. Assume that $\sum_{i \in \mathcal{I}} x_i < \frac{1}{3}$, then $\arg\min_{i \in \bar{\mathcal{I}}} x_i < \frac{1}{3}$ and the algorithm does not return yet, so at the end $\sum_{i \in \mathcal{I}} x_i \in [\frac{1}{3}, \frac{2}{3}]$. Let $p = \max\{\sum_{i \in \mathcal{I}} x_i, 1 - \sum_{i \in \mathcal{I}} x_i\}$, then $p \in [\frac{1}{3}, \frac{2}{3}]$ and the quantity in question is bounded by

$$\sum_{i=1}^{K} x_i \ell_i^2 = \sum_{i \in \mathcal{I}} x_i \ell_i^2 + \sum_{i \in \bar{\mathcal{I}}} x_i \ell_i^2 = p + (1 - p)\left(\frac{p}{1-p}\right)^2 = \frac{p}{1-p} \geq \frac{1}{2}.$$

∎

**Claim 19.** *For the negentropy potential $F(x) = \eta^{-1} \sum_{i=1}^{K} \log(x_i)x_i$, it holds that*

$$-\overline{F}^*(-L) - \min_i L_i = \eta^{-1}\log(\max_i \nabla \overline{F}^*(-L)_i).$$

*Proof.* Denote $i^* = \operatorname{argmin}_{i \in [K]} L_i$. It is well known that the exponential weights distribution is $(\nabla \overline{F}^*(-L))_i = \exp(-\eta L_i)/(\sum_{j \in [K]}) \exp(-\eta L_j)$. Therefore, the negentropy has an explicit form of the constrained convex conjugate:

$$\overline{F}^*(-L) = \left\langle \nabla \overline{F}^*(-L), -L \right\rangle - F(\nabla \overline{F}^*(-L)) = \eta^{-1} \log(\sum_{i=1}^{K} \exp(-\eta L_i)).$$

Hence

$$-\overline{F}^*(-L) - L_{i^*} = -\eta^{-1} \log \left( \sum_{i=1}^{K} \exp(-\eta L_i) \right) + \eta^{-1} \log(\exp(-\eta L_{i^*}))$$

$$= -\eta^{-1} \log \left( \frac{\sum_{i=1}^{K} \exp(-\eta L_i)}{\exp(-\eta L_{i^*})} \right) = \eta^{-1} \log \left( \nabla \overline{F}^*(-L)_{i^*} \right).$$

∎

*Proof of Theorem 9.* For ease of presentation, we will work with loss ranges $[-L_t/2, L_t/2]$, which is equivalent to loss ranges of $[0, L_t]$ in full-information games. Assume that

$$\frac{1}{2} \sum_{t=1}^{\lfloor \log_2(K) \rfloor} L_t \geq \frac{1}{32} \sqrt{\sum_{t=\lfloor \log_2(K) \rfloor}^{T} L_t^2 \log(K)}.$$

Define the active set $\mathcal{A}_1 = [K]$. At any time $t$, if $\rho(t) \notin [\lfloor \log_2(K) \rfloor]$, we set $\ell_t$ to 0 and proceed with $\mathcal{A}_{t+1} = \mathcal{A}_t$. Otherwise, if $\rho(t) \in [\lfloor \log_2(K) \rfloor]$, we randomly select half of the arms in $\mathcal{A}_t$ to assign $\ell_{t,i} = -L_{\rho(t)}/2$, and the other half $\ell_{t,i} = L_{\rho(t)}/2$. (In case of an uneven number $|\mathcal{A}_t|$ we leave one arm at 0.) All other losses are 0. We reduce $\mathcal{A}_{t+1} = \{i \in \mathcal{A}_t \mid \ell_{t,i} < 0\}$ to the set of arms that were negative. The set $\mathcal{A}_n$ will not be empty since we can repeat halving the action set exactly $\lfloor \log_2(K) \rfloor$ many times. The expected loss of any player is always 0, while the loss of the best arm is $\min_a \sum_{t=1}^{T} \ell_{t,a} = -\sum_{t=1}^{\lfloor \log_2(K) \rfloor} L_t/2$, hence

$$\mathbb{R}^* \geq \sum_{t=1}^{\lfloor \log_2(K) \rfloor} L_t/2.$$

It remains to analyse the case

$$\frac{1}{2} \sum_{t=1}^{\lfloor \log_2(K) \rfloor} L_t < \frac{1}{32} \sqrt{\sum_{t=\lfloor \log_2(K) \rfloor}^{T} L_t^2 \log(K)}.$$

In this case, note that we have

$$\sqrt{\sum_{t=\lfloor \log_2(K) \rfloor}^{T} L_t^2 / \log(K)} > \frac{16}{\log(K)} \sum_{t=1}^{\lfloor \log_2(K) \rfloor} L_t > 16 \frac{\lfloor \log_2(K) \rfloor}{\log(K)} L_{\lfloor \log_2(K) \rfloor} > 8 L_{\lfloor \log_2(K) \rfloor}. \qquad (46)$$

The high level idea is now to create a sequence of losses adapted to the choices of the algorithm. Let $x_{ti} = \mathbb{E}[I_t = i | \ell_{t-1}, \ldots, \ell_1]$ be the expected trajectory of the algorithm and let $z_{ti} = \exp(-\eta L_{ti})/\sum_{j=1}^{K} \exp(-\eta L_{tj})$ for $L_t = \sum_{s=1}^{t-1} \ell_t$ be the trajectory of EXP3. Let the adversary follow Algorithm 3 for the selection of losses, then based on Lemma 18 we have $0 = \langle z_t, \ell_t \rangle$ and also it is easy to see $0 \leq \langle x_t, \ell_t \rangle$, therefore we have $0 = \langle z_t, \ell_t \rangle \leq \langle x_t, \ell_t \rangle$. This implies that the regret of the algorithm cannot be smaller than that of EXP3, so the regret of algorithm $\mathcal{A}$ can be bounded as

$$Reg_T(\mathcal{A}) = \sum_{t=1}^{T} \langle x_t, \ell_t \rangle - \min_{a^* \in \Delta([K])} \langle a^*, L_{T+1} \rangle \geq \sum_{t=1}^{T} \langle z_t, \ell_t \rangle - \min_{a^* \in \Delta([K])} \langle a^*, L_{T+1} \rangle = -\min_{a^* \in \Delta([K])} \langle a^*, L_{T+1} \rangle ,$$

Let $F(x) = \eta^{-1} \sum_{i=1}^{K} x_i \log(x_i)$ then we have

$$-\min_{a^* \in \Delta([K])} \langle a^*, L_{T+1} \rangle = \sum_{t=1}^{T} \left[ \overline{F}^*(-L_{t+1}) - \overline{F}^*(-L_t) \right] + \overline{F}^*(-L_1) - \overline{F}^*(-L_{T+1}) - \min_{a^* \in \Delta([K])} \langle a^*, L_{T+1} \rangle$$

$$= \sum_{t=1}^{T} \eta^{-1} \log \left( \sum_{i=1}^{K} \exp(-\eta L_{t+1,i}) \right) - \eta^{-1} \log \left( \sum_{i=1}^{K} \exp(-\eta L_{t,i}) \right)$$

$$+ \eta^{-1} \log(K) + \eta^{-1} \log(\max_{i \in [K]} z_{T+1,i})$$

$$= \sum_{t=1}^{T} \eta^{-1} \log \left( \sum_{i=1}^{K} z_{ti} \exp(-\eta \ell_{ti}) \right) + \eta^{-1} \log(K) + \eta^{-1} \log(\max_{i \in [K]} z_{T+1,i}),$$

where the second equality uses Claim 19 for $L = L_{T+1}$ and the fact for any $L \in \mathbb{R}^K$, $\overline{F}^*(-L) = \eta^{-1} exp(\sum_{i=1}^K -\eta L_i)$. Now we choose the learning rate for EXP3 to be $\eta = \sqrt{\log(K)/(\sum_{t=\lfloor \log_2(K) \rfloor}^T L_t^2)}$, that based on (46) together with the fact that we set losses to zero for $\rho(t) \in [\lfloor \log_2 K \rfloor]$ in Algorithm 3 ensures $|\eta \ell_{ti}| \leq \frac{1}{2}\eta L_{\lfloor \log_2(K) \rfloor} \leq \frac{1}{2}$. Using that, by Taylor's theorem and the monotonicity of the second derivative of exp, we have for all $x \geq -\frac{1}{2}$: $\exp(x) \geq 1 + x + \frac{1}{2}\exp''(-\frac{1}{2})x^2 \geq 1 + x + \frac{3}{10}x^2$, as well as by concavity of log for all $0 \leq x \leq \frac{1}{4}$ we have $\log(1+x) \geq 4\log(5/4)x \geq \frac{5}{6}x$, we get for any $t \in [T]$ by Lemma 18

$$\eta^{-1}\log(\sum_{i=1}^K z_{ti}\exp(-\eta\ell_{ti})) \geq \eta^{-1}\log(1 + \eta^2\frac{3}{10}\sum_{i=1}^K z_{ti}\ell_{ti}^2) \geq \frac{\eta}{4}\sum_{i=1}^K z_{ti}\ell_{ti}^2 \geq \mathbb{I}\{\max_i z_{ti} \leq \frac{2}{3}\}\frac{\eta}{32}L_{\rho^{-1}(t)}^2.$$

Now we have two possible events, either $\forall t \in [T] : \max_i z_{ti} \leq \frac{2}{3}$ and

$$Reg_T(\mathcal{A}) \geq \frac{\eta}{32}\sum_{t=\lfloor \log_2(K) \rfloor} L_t^2 = \frac{1}{32}\sqrt{\sum_{t=\lfloor \log_2(K) \rfloor} L_t^2 \log(K)},$$

or there exists $s \in [T]$ such that $\max_i z_{s,i} > \frac{2}{3}$, then from Algorithm 3 we infer that $\forall t \geq s : \ell_t = 0$ and consequently $\forall t \geq s : z_t = z_s$, so $\max_i z_{T+1,i} > \frac{2}{3}$ and

$$Reg_T(\mathcal{A}) \geq \eta^{-1}(\log(K) + \log(2/3)) \geq \frac{1}{32}\eta^{-1}\log(K) = \frac{1}{32}\sqrt{\sum_{t=\lfloor \log_2(K) \rfloor} L_t^2 \log(K)}.$$

∎

---

**Algorithm 3:** Adversary

**Input:** Actor $\mathcal{A}$, learning rate $\eta$

1 **for** $t = 1, \ldots, n$ **do**

2      Set $\forall i : z_{ti} = \exp(-\eta L_{ti})/\sum_{j=1}^K \exp(-\eta L_{tj})$

3      **if** $\max_{i \in [K]} z_{ti} > \frac{2}{3}$ *or* $\rho(t) \leq \lfloor \log_2(K) \rfloor$ **then**

4          $\ell_t = 0$

5      **else**

6          Get $\ell$ from Algorithm 2 with $x = z_t$.

7          Determine $x_t = \mathbb{E}\left[\mathcal{A}((\ell_s)_{s=1}^{t-1})\right]$

8          Set $\ell_t = \text{sign}(\langle x_t, \ell \rangle)L_{\rho^{-1}(t)}\ell/2$