# OpenReview forum: "A Best-of-Both-Worlds Algorithm for Bandits with Delayed Feedback"
_NeurIPS.cc/2022/Conference — NeurIPS 2022 Accept_

### Official Review · Reviewer_BKkg · 2022-07-09

**Rating:** 7
**Confidence:** 3
**Soundness:** 3 good
**Presentation:** 3 good
**Contribution:** 3 good

**Summary:**

This study considers the multi-armed bandit problem with delayed feedback.
For this problem, the authors propose a best-of-both-worlds algorithm that achieves minimax optimal regret for adversarial regimes as well as logarithmic regret for stochastic regimes.
This result is achieved by modifying the update rules for learning rates of an existing algorithm proposed by [Zimmert and Seldin, 2020].
The proposed approach works even for problems with arbitrary (round-dependent) delays.

**Questions:**

- Are there any known lower bounds for stochastic settings with delayed feedback?

- Can the proposed algorithm handle stochastic regimes with adversarial corruption or adversarial regimes with self-bounding constraint given in [Zimmert and Seldin, 2021]?
Since the analysis is based on this technique, it appears to be easily extendable.
If not, I would like to know why.

**Limitations:**

The limitations are adequately addressed.

**Strengths And Weaknesses:**

Strengths:

- The paper makes solid contributions to important problems.
- The paper is well structured and easy to read.
- The proposed approach appears simple and practical.

Weaknesses:

- Novelty in algorithms and analysis techniques is somewhat limited.

Comments:

Delayed feedback and best-of-both-worlds algorithms are both topics of practical importance.
I consider this paper to be of high importance because it provides a solid contribution to these two topics.

The proposed algorithm and analysis techniques are based on existing ones and have limited novelty.
However, there are several nontrivial steps in the analysis that require delicate attention, which the authors address in a sophisticated manner.

---

> ### Author Response · Authors · 2022-08-02
> **Response to Reviewer BKkg**
>
> We thank the reviewer for their time and interesting questions.
>
> > Are there any known lower bounds for stochastic settings with delayed feedback?
>
> Regarding lower bounds for the stochastic setting: For uniform delays (constant delay $d$ for all rounds), a trivial lower bound is $\Omega(\sum_{i\neq i^*}\frac{\log T}{\Delta_i} + d \frac{\sum{i \neq i*} \Delta_i}{K})$. It follows from the fact that the first $d$ rounds are played ``blindly'' without any information available and in expectation the agent cannot do better than playing random actions.
> Upper bounds for algorithms tailored to the stochastic regime obtain $\mathcal{O}(\sum_{i\neq i^*}\frac{\log T}{\Delta_i} + \Delta_{\max}d )$ [Joulani and Gyorgy, 2013], which shows that the lower bound is tight within a minor factor in the time-independent lower order term. We are unaware of  instance-dependent stochastic lower bounds for varying delays. As already mentioned, this would be an interesting question for future research.
>
> > Can the proposed algorithm handle stochastic regimes with adversarial corruption or adversarial regimes with self-bounding constraint given in [Zimmert and Seldin, 2021]? Since the analysis is based on this technique, it appears to be easily extendable. If not, I would like to know why.
>
> Regarding adversarial regimes with self-bounding constraints (including corrupted regimes as a special case): Indeed, our result and analysis can be easily extended. We rely on the same self-bounding technique as Zimmert and Seldin (2021) and following their proof immediately yields $\mathcal{O}(B^{stoch}_T+\sqrt{B^{stoch}_T C})$, where $B^{stoch}_T$ is the regret upper bound in the stochastic regime proven in our paper and $C$ the total corruption budget. We will add a formal statement and a proof to the paper.

---

### Official Review · Reviewer_vDeQ · 2022-07-12

**Rating:** 7
**Confidence:** 4
**Soundness:** 4 excellent
**Presentation:** 3 good
**Contribution:** 3 good

**Summary:**

This paper present an algorithm which achieves the best of both worlds guarantee with delayed feedback between the adversarial world and stochastic world. More specifically, the authors consider two delay settings (fix delay and arbitrary uniform delay) of multi-armed bandit problem, and propose two algorithms that achieve  $\widetilde{O}(log K + K^{1/3} )$ (the ideal stochastic bound should be $\widetilde{O}(log K)$) regret within stochastic setting where the loss are sampled i.i.d from a fixed distribution, while ensuring $\widetilde{O}(\sqrt{K})$ in the worst case. The algorithms applies the FTRL framework with specific hybrid regularizers and careful learning rate scheduling, which is similar to those from previous works. However, with the novel analysis technique (Lemma 4), the authors show that the proposed algorithms actually achieve the best of both worlds.

**Questions:**

1. Is it able to achieve Lemma 4 with episodic MDPs?

2. Is it possible to improve the $d_{max} K^{1/3} \log K$ and $d K^{1/3} \log K$  terms in the regret bounds for the stochastic setting? What's the main difficulty in removing such terms from the regret bounds? Is it possible to use SAPO or EXP3++ approaches to avoid these terms?

3. Is it possible to achieve similar results with arbitrary delays with respect to the total delay $D = \sum_{t=1}^{T} d_t$?

4. Any corruption result?

**Limitations:**

None.

**Strengths And Weaknesses:**

Unlike the original multi-arm bandit problem which receives the feedback immediately after playing an arm, the delayed feedback setting is much more complicated as the learner has to maintain sufficient robustness with missing information, which may lead to drastic different strategy. Moreover, it is even harder to achieve best of both worlds results in this setting, which requires the learner to carefully balance the exploration and exploitation.

To the best of my knowledge, this is the first best of both world results of multi-armed bandit problem with the delayed feedbacks. The analysis is quite different from the approach (and "cheating-regret"-"drift" decomposition) from $Gy\ddot{o}rgy$ and Joulani (2021)  and the other approach of Zimmert and Seldin (2020), with the help of the powerful Lemma 4 which controls the drift of the playing distribution by the time-varying hybrid regularizer to handle the arbitrary delays (arbitrary uniform or fixed). Besides, the proof of Lemma 4 is very interesting and may be applied in many related problems.

This paper does not have any specific weakness point. Overall, the writing is clean and well-organized.

---

> ### Author Response · Authors · 2022-08-02
> **Response to Reviewer vDeQ**
>
> We thank the reviewer for their time and interesting open questions.
>
> >  Is it able to achieve Lemma 4 with episodic MDPs?
>
> Lemma 4 can be adapted to other forms of regularizations, including the Hybrid Regularizer used by Jin and Luo [2020] for the MDP problem. We have not checked the precise details, but we expect that adaptation could be achieved by modifying the learning rates, as we did in our paper.
>
> > Is it possible to improve the $d_{max} k^{1/3} \log K$ and $d k^{1/3} \log K$ terms in the regret bounds for the stochastic setting? What's the main difficulty in removing such terms from the regret bounds? Is it possible to use SAPO or EXP3++ approaches to avoid these terms?
>
> This is a very interesting question.
> On a technical level, these terms stem from the need to control the  drift of the player's distribution induced by unseen feedback.
> We use a worst-case bound the drift, which might be overly conservative in the stochastic regime. Perhaps a more refined approach could do better.
>
> We note that at the moment the best stochastic regret bounds for SAPO and EXP3++ without delay scale with $\log^2(T)$, whereas our regret bound scales with $\log(T)$. So, even if it would work (which we doubt), it would be an improvement of a lower order term at the cost of the dominating term.
>
> > Is it possible to achieve similar results with arbitrary delays with respect to the total delay $D$?
>
> $D$ is not a relevant quantity for characterizing the regret, neither in the adversarial setting (see our lower bound in Section 6), nor in the stochastic setting (in the case of uniform delays stochastic lower bound depends on $d$ rather than $D = dT$, for more details see our response to Reviewer Rykq regarding their question about existing lower bounds).
>
> > Any corruption result?
>
> Our result and analysis can be easily extended to the corrupted regime. We rely on the same self-bounding technique as Zimmert and Seldin (2021) and following their proof immediately yields $\mathcal{O}(B^{stoch}_T+\sqrt{B^{stoch}_T C})$, where $B^{stoch}_T$ is the regret upper bound in the stochastic regime proven in our paper and $C$ the total corruption budget. We will add a formal statement and a proof to the paper.

---

### Official Review · Reviewer_Rykq · 2022-07-13

**Rating:** 5
**Confidence:** 3
**Soundness:** 3 good
**Presentation:** 3 good
**Contribution:** 3 good

**Summary:**

This paper studied the problem of multi-armed bandits problem with delayed feedback. Most of the traditional algorithms focus on either stochastic or oblivious adversary settings, thus in the case where the setting is unknown, these algorithms can not achieve optimal pseudo-regret bound. In order to overcome this problem, this paper provided a slight modification version of Zimmert and Seldin [2020]'s algorithm (adding a constant term $\eta_0$ and $\gamma_0$ to the learning rate). By doing that, it can achieve near-optimal regret in both stochastic and oblivious adversary settings.

**Questions:**

Can the authors comment on the existing lower bound regret in the stochastic setting if they exist one?

**Limitations:**

There is no potential negative societal impact.

**Strengths And Weaknesses:**

I like the overall idea of the paper to extend the best-of-both-worlds algorithm into a delayed feedback setting. However, I have a few concerns over the contribution of this paper.

As the algorithm is very similar to Zimmert and Seldin [2020]'s algorithm, the first regret bound analysis against oblivious adversary is also very similar to the analysis in Zimmert and Seldin [2020]'s paper. Furthermore, due to the modification, the newly derived regret suffers another term, $d_{max} K^{1/3} \log(K)$ compared to the regret of the original algorithm.

The second regret bound against stochastic setting is interesting and novel as it provides a new feature for the Zimmert and Seldin [2020]'s algorithm (it also requires a new technique of controlling drift of the playing distribution over arbitrary time intervals).

The third contribution about adversarial regret lower bound, although novel is in my opinion not well connected to current paper since Algorithm 1 does not consider skipping-based refined regret or achieves any regret bound that matches the new lower bound.

Therefore, the main contribution of the paper is to provide a regret bound for a modification of Zimmert and Seldin [2020]'s algorithm against a stochastic setting with a small sacrifice in the regret against an oblivious adversary.  Even though I think this is new and novel, I am unsure whether it is significant enough for publication in Neurips.

minors: There are a few terms mentioned in the paper that I can not find the definition (e.g., uniform delays, minimax regret).

---

> ### Author Response · Authors · 2022-08-02
> **Response to Reviewer Rykq**
>
> We thank the reviewer for their time and feedback.
>
> We would like to say a few words regarding the significance of our contribution.
>
> 1. We would like to emphasize that, to the best of our knowledge, this is the first best-of-both-worlds result for bandits with delayed feedback and that it resolves an open question by Zimmert and Seldin [2020].
>
> 2. We would like to emphasize that control of the drift of the playing distribution in Lemma 4 is novel and highly non-trivial, as also recognized by other reviewers.
>
> 3. While the connection between the refined adversarial lower bound in Section 6 and the rest of the paper is indeed a bit loose at the moment, the lower bound is significant for several reasons. First, it establishes optimality of the skipping technique of Zimmert and Seldin for the adversarial regime with arbitrary delays. Second, we expect that skipping may eventually be used to eliminate the need in prior knowledge of the maximal delay $d_{max}$, although for now the control of the drift is already challenging enough and so far we were unable to get rid of this assumption. But if it succeeds at some point, the connection between the lower bound and the other results will be stronger. And finally, the trade-off for optimal skipping in the stochastic regime seem to be different from the trade-off for optimal skipping in the adversarial regime. Therefore, we conjecture that there is no skipping scheme that would be optimal for the adversarial and stochastic regime simultaneously, although we do not have a best-of-both-worlds lower bound for arbitrary delays yet. But this is another direction for future research that will likely strengthen the connection between our lower bound and the rest of the paper.
>
> > minors: There are a few terms mentioned in the paper that I can not find the definition (e.g., uniform delays, minimax regret).
>
> Minimax regret is the minimum of all worst case (maximum) outcomes of regret (depending on the adversary randomization). Uniform delays refers to having fixed delay $d$ in all rounds. We will add these definitions to the paper.
>
> > Can the authors comment on the existing lower bound regret in the stochastic setting if they exist one?
>
>  For uniform delays (constant delay $d$ for all rounds), a trivial lower bound is $\Omega(\sum_{i\neq i^*}\frac{\log T}{\Delta_i} + d \frac{\sum{i \neq i*} \Delta_i}{K})$. It follows from the fact that the first $d$ rounds are played ``blindly'' without any information available and in expectation the agent cannot do better than playing random actions.
> Upper bounds for algorithms tailored to the stochastic regime obtain $\mathcal{O}(\sum_{i\neq i^*}\frac{\log T}{\Delta_i} + \Delta_{\max}d )$ [Joulani and Gyorgy, 2013], which shows that the lower bound is tight within a minor factor in the time-independent lower order term. We are unaware of  instance-dependent stochastic lower bounds for varying delays. As already mentioned, this would be an interesting question for future research.

---

> ### Comment · Reviewer_Rykq · 2022-08-08
> **Post-rebuttal thought**
>
> Many thanks the authors for their response.
>
> I have read through other reviewers’ comments and as I said earlier in my initial comment, I think the paper makes novel contribution into analysing the regret bound of a modified version of a known algorithm in stochastic setting. I also acknowlege that the lower bound analysis is novel, yet I still believe it disconnects with the other part of the paper as I explained in my initial response. It would be very nice if in the future, the authors can manage to use skiping technique to improve the algorithm’ analysis so that this lower bound is more relevant.
>
> Overall, the paper is sought and I recommend a borderline accept score. I keep my initial score as I believe my concerns are still valid after the rebuttal.

---

### Meta-Review · Area_Chair_u4mv · 2022-08-27

**Recommendation:** Accept
**Confidence:** Certain

**Metareview:**

The paper makes a solid technical contribution in the online learning literature, providing the first best-of-both worlds algorithm for online learning with delayed feedback. Despite building heavily on existing algorithmic ideas, the paper involves some critical technical novelties that enable their results.

**Award:**

No

---

### Decision · Program_Chairs · 2022-09-14

Accept